# The robust, high-throughput, and temporally regulated roxCre and loxCre reporting systems for genetic modifications in vivo

Mengyang Shi[1†], Jie Li[1†], Xiuxiu Liu[1], Kuo Liu[2], Lingjuan He[3], Wenjuan Pu[1], Wendong Weng[1], Shaohua Zhang[1], Huan Zhao[1], Kathy Lui[4*], Bin Zhou[1,2,5*]

[1]CAS CEMCS-CUHK Joint Laboratories, New Cornerstone Science Laboratory, State Key Laboratory of Cell Biology, CAS Center for Excellence in Molecular Cell Science, Shanghai Institute of Biochemistry and Cell Biology, Chinese Academy of Sciences, University of Chinese Academy of Sciences, Shanghai, China; [2]Key Laboratory of Systems Health Science of Zhejiang Province, School of Life Science, Hangzhou Institute for Advanced Study, University of Chinese Academy of Sciences, Hangzhou, China; [3]School of Life Sciences, Westlake University, Hangzhou, China; [4]Department of Chemical Pathology, Li Ka Shing Institute of Health Sciences, The Chinese University of Hong Kong, Prince of Wales Hospital, Hong Kong, China; [5]School of Life Science and Technology, ShanghaiTech University, Shanghai, China

*For correspondence:
kathyolui@cuhk.edu.hk (KL);
bzhousibs@163.com (BZ)

†These authors contributed equally to this work

Competing interest: The authors declare that no competing interests exist.

## eLife Assessment

This study presents an **important** set of new tools to facilitate Cre or Dre-mediated recombination in mice. The characterization of these new tools was done using **solid** and validated methodology. The work **convincingly** demonstrates the efficient gene knockout capability of these models and will progress the field.

**Abstract** Cre-loxP technology, a cornerstone in fate mapping and in vivo gene function studies, faces challenges in achieving precise and efficient conditional mutagenesis through inducible systems. This study introduces two innovative genetic tools designed to overcome these limitations. The first, roxCre, enables DreER-mediated Cre release, paving the way for intersectional genetic manipulation that permits increased precision and efficiency. The second, loxCre, facilitates conditional gene targeting by allowing CreER lines to induce Cre expression with significantly enhanced efficiency. These tools incorporate a fluorescent reporter for genetic lineage tracing, simultaneously revealing efficient gene knockout in cells marked by the reporter. These strategies hold great potential for precise and efficient exploration of lineage-specific gene functions, marking a significant advancement in genetic research methodologies.

## Introduction

Genetic modification, such as gene knockout in a precise spatiotemporal manner, forms the basis for understanding cellular and molecular mechanisms in multiple biological processes (*Gu et al., 1993*; *Gu et al., 1994*). The Cre-loxP recombination system is widely employed to knock out or overexpress specific genes in vivo (*Nagy, 2000*). Cre recombinase, derived from the bacteriophage P1 gene,

targets short palindrome DNA sequence loxP (34 bp) and recombines codirectional-loxP-flanked sequences for deletion (*Sternberg et al., 1981*; *Sauer and Henderson, 1988*). To date, most of the mouse genes have been successfully targeted by two loxP sites (floxed allele), which could be recombined by cell-type-specific Cre lines. Application of the Cre-loxP system in mouse genetics revolutionizes gene functional analysis and significantly advances our exploration of many developmental and pathophysiological processes at the molecular level (*Orban et al., 1992*; *Lakso et al., 1992*).

While constitutively active Cre is robust and efficient in recombination for gene deletion, Cre lacks temporal control as the promoter driving Cre is active from the embryonic to the adult stage. Therefore, a temporally active Cre is needed to avoid early mortality and realize the ablation of an indispensable gene at a key developmental stage. To overcome the temporal limitation of Cre, CreER, the fusion of Cre with a mutated estrogen receptor, is generated to enable its activation only after tamoxifen (Tam) treatment, and the interaction of ER with Tam induces nuclear translocation of CreER from the cytoplasm (*Feil et al., 1997*). As a result, Tam treatment restricts CreER activity at a specific time window and permits gene functional analysis at a higher spatiotemporal resolution than Cre (*Tian et al., 2015*). As the loxP-flanked sequences and their locations in genome or chromatin influence its accessibility for recombination, the recombination efficiency of CreER varies significantly among multiple floxed alleles, ranging from easily recombined to recombination-inert alleles (*Tian et al., 2020*). For example, CreER has high efficiency in targeting the easy-to-recombine alleles, such as generic reporter *Rosa26-tdT* (*Madisen et al., 2010*), but exhibits low efficiency on some other reporters such as *Rosa26-Confetti* (*Snippert et al., 2010*), or may not excise some floxed gene alleles that are inert or resistant to targeting, leading to failure of gene deletion in some tdT$^+$ cells and causing false-positive tracing fate (*Figure 1A*). In other words, tdT expression may not always necessarily denote gene deletion in the labeled cells. The notion that the pattern of Cre expression based on a reporter (e.g. *Rosa26-tdT*) represents the pattern of Cre-mediated recombination is less rigorous (*Becher et al., 2018*). Therefore, it is not precise to assume the successful deletion of codirectional-loxP-flanked sequences in the gene of interest based on the deletion in another gene (or reporter) by the same CreER mouse line.

One way to increase the recombination efficiency of CreER is to treat mice with many times of Tam, as the high dosage of Tam in theory could increase the chances of CreER-mediated recombination. However, the increased dosage of Tam treatment is toxic and could lead to many side effects on the phenotype, creating confounding effects on the study (*Brash et al., 2020*; *Rashbrook et al., 2022*). To increase the inducible recombination efficiency of floxed alleles, Lao et al. reported a mosaic mutant analysis with spatial and temporal control of recombination (MASTR), in which initial FlpoER-mediated recombination induces GFPCre expression, that subsequently recombines floxed alleles effectively in GFP$^+$ cells (*Lao et al., 2012*). Similarly, a Flp-induced mosaic analysis system with Cre or Tomato (MASCOT) has been reported to express Cre and reporter simultaneously in a cell (*Wang et al., 2020*). While MASTR and MASCOT enable spatiotemporal labeling of mosaic mutant cells utilizing the current resources of floxed mouse lines, it may not be suitable for effective gene knockout at a tissue/population level due to low recombination efficiency initiated by Flp recombinase. In order to improve the inducible recombination efficiency, Tian et al. have also reported a self-cleaved inducible CreER (sCreER) that switches inducible CreER into a constitutively active Cre, effectively recombining floxed allele for gene manipulation (*Tian et al., 2020*). However, self-cleaved inducible Cre mice have to be newly generated each time for targeting different promoters, which is time-consuming, and the system is not compatible with current resources largely based on CreER mouse lines. A recent study has reported an iSuRe-Cre strategy to induce and report Cre-dependent genetic modifications utilizing conventional CreER tools (*Fernández-Chacón et al., 2019*). The reporter expression reflects gene knockout in cells by iSuRe-Cre. However, two limitations hinder its widespread applications. First, without any induction, the *iSuRe-Cre* transgene is leaky in some tissues, such as heart and skeletal muscle. Second, the initial recombination for releasing the *iSuRe-Cre* allele induced by CreER is far from efficient in some tissues compared with easy-to-recombine alleles such as *Rosa26-tdT* (*Fernández-Chacón et al., 2019*), limiting its usage for efficient gene knockout at a tissue level. Thus, a new method is needed to ensure gene knockout in cells as effectively and efficiently as recombination on easy-to-recombine alleles, thus allowing efficient gene deletion at a population level.

Conventional Cre-loxP system uses tissue-specific promoter to drive Cre, thus the precision of genetic targeting solely depends on promoter activity. Now it is known that many promoters are not

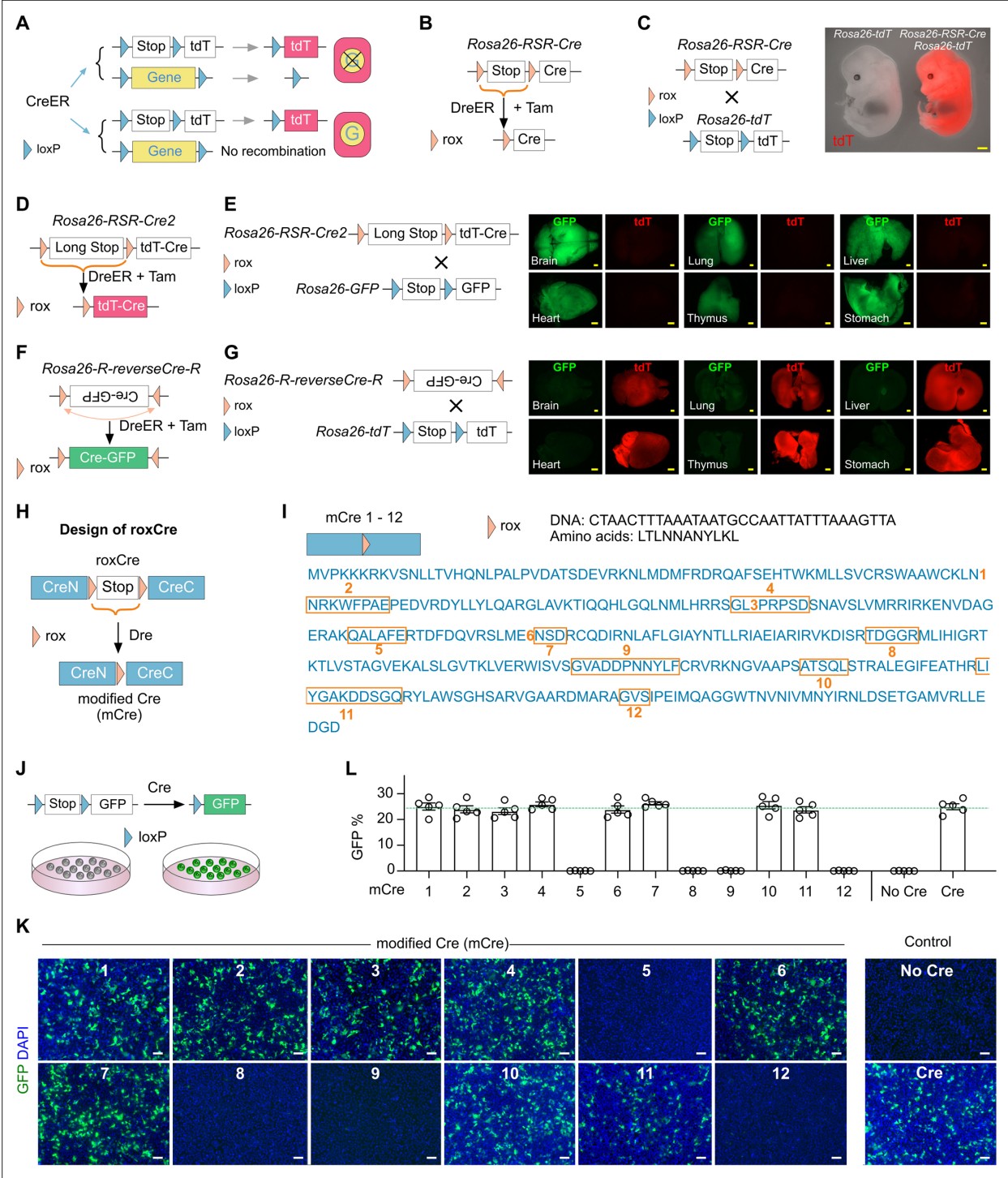

**Figure 1.** Design of roxCre for DreER-induced mCre expression. (**A**) A schematic showing genetic labeling and/or gene knockout in CreER-expressing cells. (**B, D, F**) Strategies for DreER-induced Cre expression. (**C**) Crossing with *Rosa26-tdT* mice, *Rosa26-RSR-Cre* exhibits leakiness in E13.5 embryos. Scale bars, yellow 1 mm. Each image represents 5 individual biological samples. (**E**) Examination of Cre leakiness by whole-mount fluorescence images of organs collected from *Rosa26-RSR-Cre2;Rosa26-GFP* adult mice. Scale bars, yellow 1 mm. Each image represents 5 individual biological samples. (**G**) Examination of Cre leakiness by whole-mount fluorescence images of organs collected from *Rosa26-R-reverseCre-R;Rosa26-tdT* adult mice. Scale bars, yellow 1 mm. Each image represents 5 individual biological samples. (**H**) A schematic showing the design for roxCre and mCre. (**I**) A schematic showing mCre1 to mCre12 by insertion of rox into *Cre* cDNA at 12 different loci. (**J**) Examination of mCre for recombination efficiency by the *loxP-Stop-loxP-GFP* reporter. (**K**) Immunofluorescence images of cells stained with GFP and DAPI. Scale bars, white, 100 μm. Each image represents 5 individual biological samples. (**L**) Quantification of the percentage of cells expressing GFP in each group. Data are the means ± SEM; n=5.

*Figure 1 continued on next page*

*Figure 1 continued*

The online version of this article includes the following source data for figure 1:

**Source data 1.** Numerical data to generate *Figure 1L*.

as specific as previously reported (*Liu et al., 2016*; *Gribben et al., 2023*; *Furuyama et al., 2011*). The ectopic or unwanted Cre expression in other cell types may result in unspecific cell labeling or gene deletion, leading to many confounding issues or controversies in multiple fields of research (*Xiao et al., 2013*; *Tarlow et al., 2014*; *He et al., 2020*). In addition, some cell types cannot be clearly distinguished by one marker from another, and thus could not be specifically and genetically targeted by only one gene promoter-driven recombinase (*Liu et al., 2019*). To circumvent this limit, dual recombinase-mediated genetic targeting utilizes two promoters to independently drive Cre and Dre (*He et al., 2017*). Dre is another bacteriophage recombinase that targets rox sites and is orthogonal to the Cre-loxP system (*Anastassiadis et al., 2009*). While cells of interest could be labeled by specific reporters that are responsive to dual recombinases, gene deletion requires a final readout on singular Cre recombinase for targeting floxed gene alleles.

In this study, we developed two genetic strategies: roxCre and loxCre, in which the *rox-Stop-rox* (RSR) and *loxP-Stop-loxP* cassettes are respectively inserted into the *Cre* coding region, such that *Cre* transcription and translation are not properly carried out before the removal of the *Stop* cassette. There is no obviously spontaneous leakiness by roxCre or loxCre. We found that CreER-induced loxCre effectively deletes genes in reporter-labeled cells, ensuring efficient gene knockout for the evaluation of lineage-specific gene function. In addition, DreER-induced roxCre efficiently deletes genes in cells for both specific and efficient intersectional genetic studies. We expect that these tools would overcome the issues of inefficient and nonspecific genetic modifications, enhancing the ability to more precisely manipulate genes in a particular cell lineage for a better understanding of gene functions in multiple biomedical fields.

## Results

### Design of roxCre for DreER-induced mCre expression

A sequential genetic approach has been recently reported to delete genes (*Han et al., 2023*), which used Dre-induced CreER expression by removing the *rox-Stop-rox* sequence ahead. As aforementioned, the recombination carried out by the released CreER for excising floxed gene alleles may not be as efficient as reporters (e.g. *Rosa26-tdT*). A possible strategy to control Cre activity is to engineer a rox-flanked *Stop* sequence ahead of its DNA (*Rosa26-RSR-Cre*) that will be removed after Dre-rox recombination (*Figure 1B*). However, such a strategy is not ideal, as the upstream transcriptional *Stop* sequence might not completely prevent *Cre* transcription, leading to leakiness (*Figure 1C*). We then inserted a longer *Stop* sequence or reversed the direction of the *Cre* DNA by generating *Rosa26-RSR-Cre2* and *Rosa26-R-reverseCre-R* lines, respectively. The whole-mount fluorescence imaging results displayed heavy leakiness, which demonstrates that Cre was independently activated for recombination without being crossed with a Dre mouse line (*Figure 1D–G*). To enable efficient dual recombinase-mediated gene knockout, we need to design a strategy for DreER-induced expression of a constitutively active Cre, and the system should not exhibit leakiness.

We first generated roxCre, in which an RSR cassette was inserted into the coding region of *Cre*, splitting Cre into N- and C-terminal segments (*Figure 1H*). After removal of RSR by Dre-rox recombination, Cre is recombined, containing one remaining rox site within its coding sequence, hereafter termed modified Cre (mCre, *Figure 1H*). To ensure that mCre has the same recombination efficiency as the conventional Cre, we inserted a rox sequence at 12 different sites of *Cre* individually, most of which were located between helix domains (*Figure 1I*). As insertions of additional amino acids of rox would change the original sequence of Cre, potentially leading to reduced activity, we screened for their recombination efficiency by transfecting cells with 12 versions of mCre (mCre1 to mCre12) and the responding GFP reporter (*Figure 1J*). We found that mCre1, 2, 3, 4, 6, 7, 10, and 11 displayed a comparable efficiency to the conventional Cre, whereas mCre5, 8, 9, and 12 exhibited impaired recombination efficiency (*Figure 1K and L*). These data demonstrate that some of the mCre generated in our study are as efficient as the conventional Cre in vitro.

Next, we sought to screen for the most robust version of mCre in vivo. We generated four roxCre knock-in mice based on mCre1, 4, 7, and 10 as determined from the above in vitro screening (*Figure 2—figure supplements 1 and 2*). In these roxCre lines, RSR inserted into *Cre* was removed by Dre-rox recombination, resulting in the generation of mCre (*Figure 1H*). We then used hepatocyte- and endothelial cell-specific promoters albumin (Alb) and VE-cadherin (Cdh5) to drive roxCre and generate *Alb-roxCre1-tdT*, *Cdh5-roxCre4-tdT*, *Alb-roxCre7-GFP*, and *Cdh5-roxCre10-GFP* knock-in mice, respectively. No spontaneous fluorescence signal was detected in *Alb-roxCre1-tdT*, *Cdh5-roxCre4-tdT*, *Alb-roxCre7-GFP*, and *Cdh5-roxCre10-GFP* knock-in mice, demonstrating no reporter leakiness without Dre-rox recombination. Nor did we detect leakiness as a result of any Cre activity when they were crossed with *Rosa26-GFP* (*Zhang et al., 2016a*) or *Rosa26-tdT* (*Madisen et al., 2010*) reporter (*Figure 2—figure supplements 1 and 2*). These data demonstrate that roxCre mouse lines are functionally efficient and non-leaky.

## DreER-induced mCre robustly recombines inert alleles

To more rigorously evaluate the recombination effectiveness among different versions of mCre, we used an inducible DreER to temporally and specifically release mCre in hepatocytes or endothelial cells. Generic reporter mouse lines *Rosa26-GFP* and *Rosa26-tdT* were used to indicate the difference in recombination efficiency between roxCre1 and roxCre7 first. The fluorescence-activated cell sorting (FACS) and immunostaining results showed no leakiness in *Rosa26-DreER;Alb-roxCre1-tdT;Rosa26-GFP* and *Rosa26-DreER;Alb-roxCre7-GFP;Rosa26-tdT* (*Figure 2—figure supplement 3C–F*). Nevertheless, both versions of mCre efficiently labeled all targeted cells (*Figure 2—figure supplement 3C–E, G*) due to the easy-to-recombine characteristic of *Rosa26-GFP* and *Rosa26-tdT*.

To evaluate a strong mCre, we attempted to target one of the most inert alleles for recombination, *Rosa26-Confetti* (*Snippert et al., 2010*), which is often used as reporters for clonal analysis due to its sparse labeling with rare recombination events (*Tian et al., 2020*; *Snippert et al., 2010*). According to the Cre-loxP recombination principle, Cre can remove the sequence between two loxP sites that have the same transcription direction and turn over the sequence in the middle of two loxP sites that have the opposite transcription direction. As the recombination efficiency of CreER is limited, *Rosa26-Confetti* can only be recombined into YFP, or nGFP (nuclear GFP), or mCFP (membrane CFP), or RFP (*Figure 2—figure supplement 4A*).

We crossed *Rosa26-DreER;Rosa26-Confetti* mice with *Alb-roxCre1-tdT* (group 1) and *Alb-roxCre7-GFP* (group 2) mice, and Tam-induced Dre-rox recombination yielded *Alb-mCre1-tdT* and *Alb-mCre7-GFP* mice, respectively (*Figure 2A*). In this system, the constitutively active mCre would completely remove the sequence between two loxP sites that have the same transcription direction and recombine *Rosa26-Confetti* into two sets of reporter pairs: YFP-nGFP pair and mCFP-RFP pair, which could be detected in a single hepatocyte due to the poly-nuclear or polyploidy feature of hepatocytes (*Figure 2A*). In each pair, reporters can equally express, as mCre is strong enough to turn over the sequence in the middle of two loxP sites that have the opposite transcription direction (*Figure 2—figure supplement 4B*). Therefore, we used YFP/mCFP to detect mCre recombination efficiency and tdT/RFP or GFP to trace its endogenous reporter activity that was driven by Alb promoter in *Alb-mCre1-tdT* and *Alb-mCre7-GFP* mice, respectively (*Figure 2A*). Besides, *Alb-CreER;Rosa26-Confetti* (*He et al., 2017*) was used as a control under the same Tam treatment (group 3, *Figure 2A*).

In fact, DreER-rox recombination in hepatocytes is not 100%. Therefore, we quantified YFP and mCFP expression in hepatocytes with DreER-rox recombination to evaluate the strength of mCre activity. We found sparse YFP$^+$ or mCFP$^+$ hepatocytes (0.39 ± 0.04%) in *Alb-CreER;Rosa26-Confetti* mice. However, 99.94 ± 0.02% of tdT$^+$ hepatocytes (DreER recombined) were YFP$^+$ or mCFP$^+$ in *Rosa26-DreER;Alb-roxCre1-tdT;Rosa26-Confetti* mice; and 76.83 ± 3.20% of GFP$^+$ hepatocytes (DreER recombined) were YFP$^+$ or mCFP$^+$ in *Rosa26-DreER;Alb-roxCre7-GFP;Rosa26-Confetti* mice (*Figure 2B and C*). Similarly, we found that 39.81 ± 2.61% of tdT$^+$ endothelial cells and 74.15 ± 3.64% of GFP$^+$ endothelial cells in small intestine were YFP$^+$ or mCFP$^+$ in *Rosa26-DreER;Cdh5-roxCre4-tdT;Rosa26-Confetti* and *Rosa26-DreER;Cdh5-roxCre10-GFP;Rosa26-Confetti* mice, respectively, compared with sparse labeling in *Cdh5-CreER;Rosa26-Confetti* mice (*Figure 2D–F*). Similar results were also observed in other tissues and organs (*Figure 2—figure supplement 5*). Furthermore, *Rosa26-DreER;Alb-roxCre1-tdT;Rosa26-Confetti*, *Rosa26-DreER;Alb-roxCre7-GFP;Rosa26-Confetti*, *Rosa26-DreER;Cdh5-roxCre4-tdT;Rosa26-Confetti*, and *Rosa26-DreER;Cdh5-roxCre10-GFP;Rosa26-Confetti*

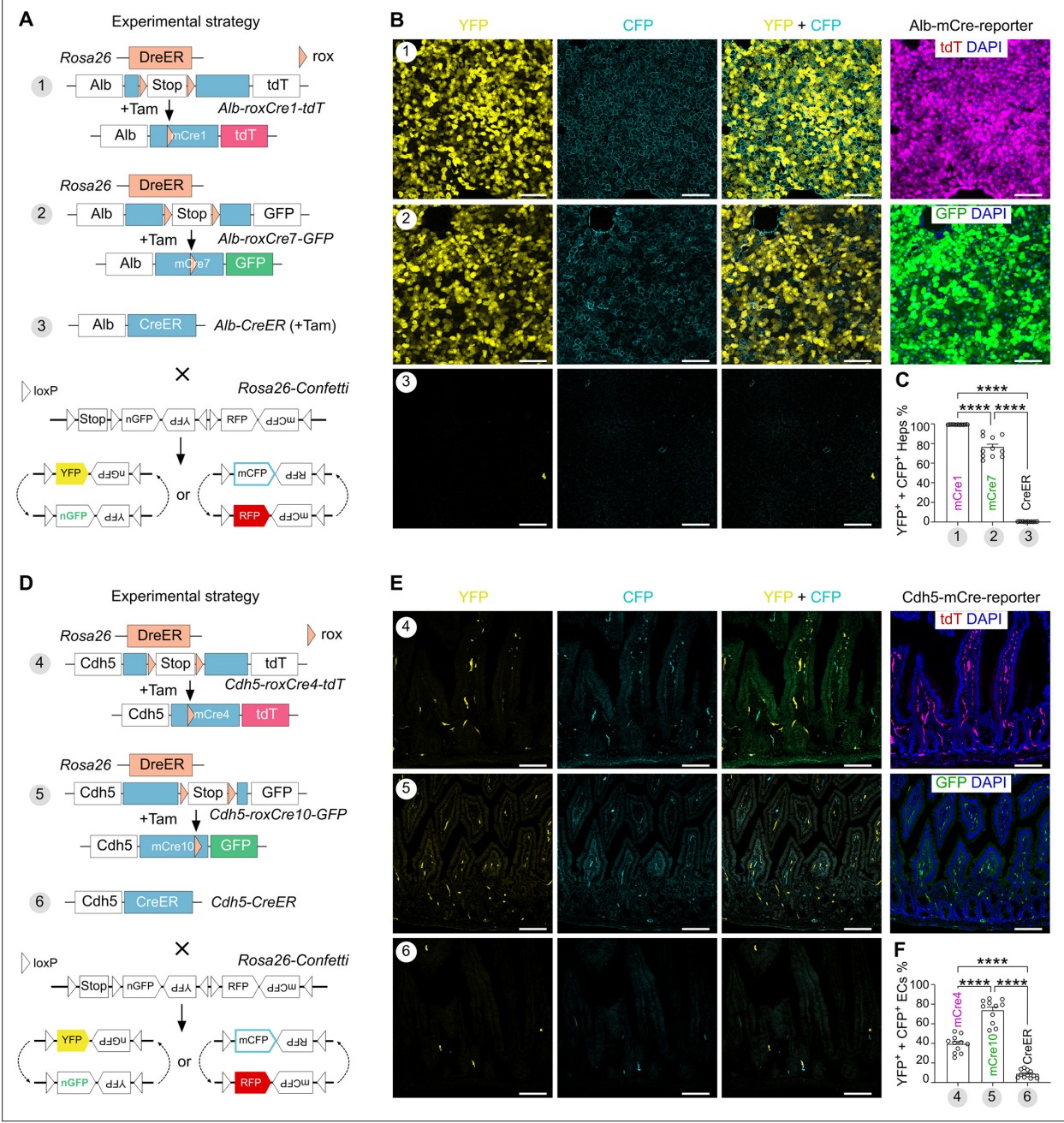

**Figure 2.** DreER-induced mCre robustly recombines inert alleles. (**A**) A schematic showing the experimental design to test the recombination efficiency of mCre1 and mCre7 on the *Rosa26-Confetti* allele. Tam-induced DreER-rox recombination leads to mCre1/tdT or mCre7/GFP expression in hepatocytes in Strategy 1 or 2, respectively. Strategy 3 uses conventional *Alb-CreER* as control. YFP and mCFP signals are used for the examination of recombination on the *Rosa26-Confetti* allele. (**B**) Fluorescence images of YFP and mCFP on liver sections collected from mice in Strategies 1–3. Strategies 1 and 2 exhibit tdT or GFP in hepatocytes after DreER-rox recombination, respectively. Scale bars, 100 μm. Each image is representative of 11 individual biological samples. (**C**) Quantification of the percentage of hepatocytes (Heps) expressing either YFP and/or mCFP in Strategies 1–3. Data are the means ± SEM; n=11 mice for each strategy; ****p<0.0001 by one-way ANOVA. (**D**) A schematic showing the experimental strategy to test the recombination efficiency of mCre4 and mCre10 on *Rosa26-Confetti* allele. Tam-induced recombination leads to mCre4/tdT or mCre10/GFP expression in endothelial cells (ECs) in Strategy 4 or 5, respectively. Strategy 6 uses conventional *Cdh5-CreER* as control. YFP and mCFP signals are used for the examination of recombination on *Rosa26-Confetti* allele. (**E**) Fluorescence images of YFP and mCFP on intestinal sections collected from mice in Strategies 4–6. Strategies 4 and 5 exhibit tdT or GFP in ECs after DreER-rox recombination, respectively. Scale bars, 100 μm. Each image is representative of 11 individual biological samples. (**F**) Quantification of the percentage of ECs expressing either YFP and/or mCFP in Strategies 4–6. Data are the means ± SEM; n=11 mice for each strategy; ****p<0.0001 by one-way ANOVA.

The online version of this article includes the following source data and figure supplement(s) for figure 2:

*Figure 2 continued on next page*

**Figure supplement 1.** No leakiness in *Alb-roxCre1-tdT;Rosa26-GFP* and *Cdh5-roxCre4-tdT;Rosa26-GFP* mice.

**Figure supplement 2.** No leakiness in *Alb-roxCre7-GFP;Rosa26-tdT* and *Cdh5-roxCre10-GFP;Rosa26-tdT* mice.

**Figure supplement 3.** Limitations of readily recombined reporters in assessing mCre-loxP recombination efficiency.

**Figure supplement 4.** Cre-loxP recombination efficiency among individual mice.

**Figure supplement 5.** Comparison of recombination efficiency mediated by mCre4 and mCre10.

were not leaky (*Figure 2—figure supplement 4C*). Together, these data demonstrate that inducible DreER controls roxCre activation, and mCre1 is the most efficient recombinase assisting the recombination of inert alleles in vivo. Hereafter, roxCre1 will be referred to as roxCre for brevity.

## DreER-induced mCre realizes cell subpopulation-specific gene manipulation

To substantiate the precision and specificity of roxCre targeting cells, we employed intersectional genetic targeting using dual recombinases. We generated the *Cyp2e1-DreER* mouse line, which labeled peri-central hepatocytes and also renal epithelial cells (*Figure 3—figure supplement 1*). Crossing *Cyp2e1-DreER* with hepatocyte-specific *Alb-roxCre-tdT* mice achieved genetic targeting of peri-central hepatocytes exclusively, circumventing issues associated with ectopic targeting of renal cells by *Cyp2e1-DreER* line or unwanted targeting of peri-portal hepatocytes by *Alb-CreER* line (*He et al., 2017*). Wnt-β-catenin plays an important role in liver development, zonation, homeostasis, and diseases (*Perugorria et al., 2019*). Therefore, we generated the *Alb-roxCre-tdT;Ctnnb1 flox/flox* mouse line to examine if Dre-induced mCre could efficiently delete *Ctnnb1* gene that encodes β-catenin (*Ctnnb1 flox*) (*Huelsken et al., 2001*).

We generated *Cyp2e1-DreER;Alb-roxCre-tdT;Ctnnb1 flox/flox* triple knock-in mice (mutant), in which Tam-induced DreER-rox recombination yielded mCre that subsequently targeted the floxed *Ctnnb1* allele (*Figure 3A*). We treated mutant mice and their littermate control *Cyp2e1-DreER;Alb-roxCre-tdT;Ctnnb1 flox/+* mice with Tam at 8 weeks of age and sorted tdT⁺ hepatocytes for analysis at 3 days after Tam treatment (*Figure 3B*). qRT-PCR analysis showed significantly reduced *Ctnnb1* and *Glul* in tdT⁺ hepatocytes of the mutant group, compared to those of the control group (*Figure 3C and D*). We then collected liver samples at 3 days (D) and 4 weeks (W) post-Tam for further analysis (*Figure 3E*). Immunostaining for tdT, β-catenin, GS, and E-CAD on liver sections revealed similar levels of β-catenin and GS expression in the 3D mutant liver compared to the control liver (*Figure 3F*). However, expression of β-catenin and GS was reduced in the 4W mutant liver tdT⁺ region compared with the control (*Figure 3G*), suggesting efficient deletion of *Ctnnb1* that participates in regulating Wnt signaling downstream gene target *Glul*, which encodes GS. Western blotting against β-catenin and qRT-PCR against *Ctnnb1* of sorted tdT⁺ hepatocytes at 4W post-Tam revealed the significant *Ctnnb1* knockout (*Figure 3H–K*). Additionally, Wnt downstream or related zonation-regulated genes, including *Glul, Axin2, Cyp1a2, Cyp2e1, Oat, Tcf7, Lect2, Tbx3, Slc1a2,* and *Rhbg,* were significantly reduced in the mutant sorted tdT⁺ hepatocytes than in the control group (*Figure 3L*). Taken together, these data demonstrate that DreER-induced mCre enables efficient intersectional genetic manipulation in the cells of interest.

## *Rosa26-loxCre-tdT* is efficiently recombined by CreER

Having successfully constructed roxCre for Dre-induced mCre expression, we iterated the system to enable mCre induction by CreER, a more widely available tool for biomedical research. We replaced the rox sites of roxCre1 with loxP sites by inserting a *loxP-Stop-loxP* cassette into *Cre* sequence at the same locus as roxCre1 (*Figure 4A*). We then generated the *Rosa26-loxCre-tdT* mouse line, in which mCre and tdT would be expressed simultaneously after *Stop* removal, allowing tdT as a surrogate marker for mCre detection in the same cell (*Figure 4C*). We then designed two experiments to verify the baseline leakiness of Cre expression (*Figure 4B*) and to examine

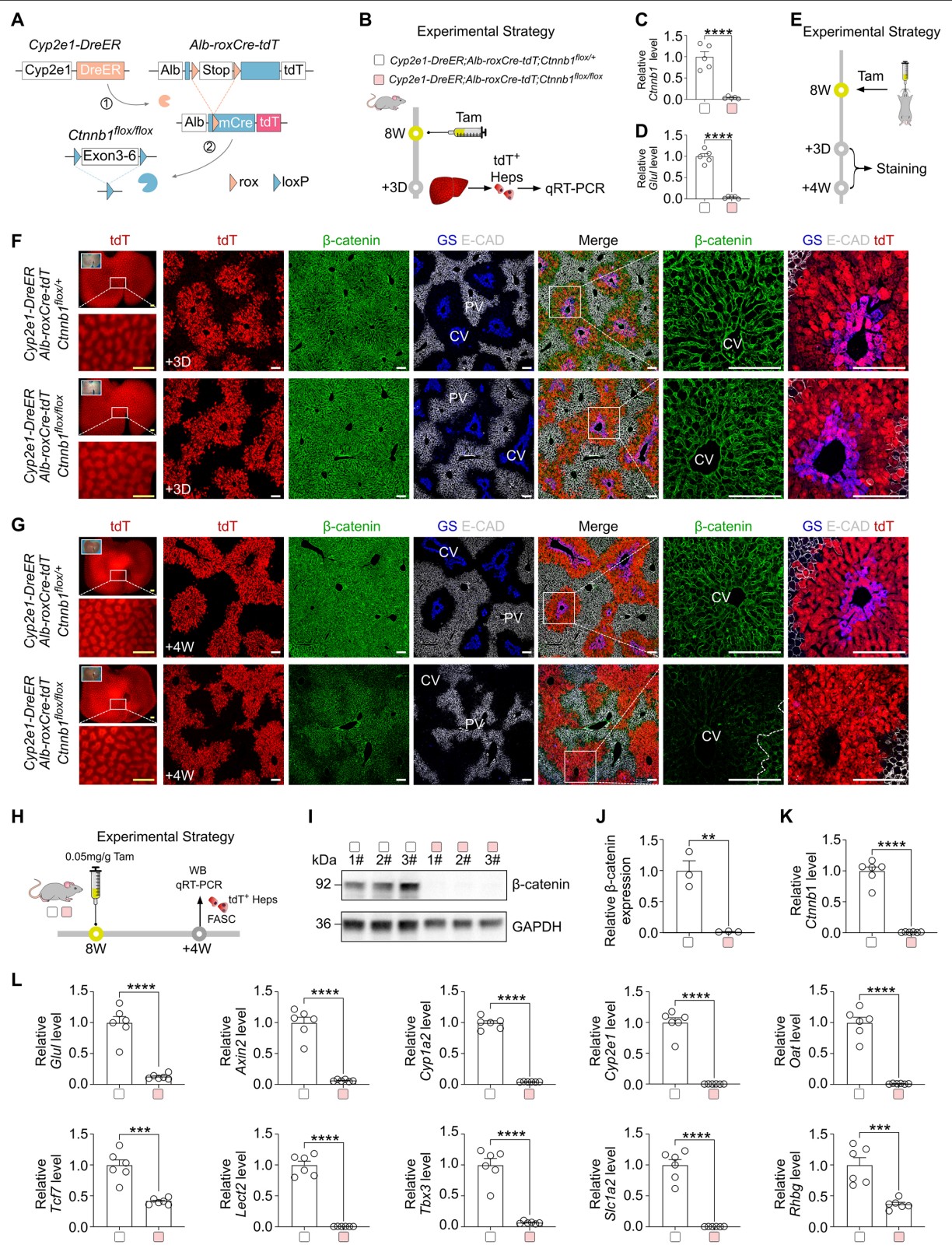

**Figure 3.** DreER-induced mCre efficiently deletes genes in specific cell subpopulations. (**A**) A schematic showing the experimental design for DreER-induced mCre expression and the subsequent gene deletion. (**B**) A schematic showing the experimental strategy. (**C and D**) qRT-PCR analysis of the relative expression of *Ctnnb1* (**C**) and *Glul* (**D**) in the sorted tdT+ hepatocytes. Data are the means ± SEM; n=5; ****p<0.0001 by Student's *t*-test. (**E**) A schematic showing the experimental strategy. (**F**) Immunostaining for tdT, β-catenin, GS, and E-CAD on liver sections collected on day 3 post-Tam. PV,

*Figure 3 continued on next page*

*Figure 3 continued*

portal vein; CV, central vein. Scale bars, yellow 1 mm; white 100 μm. Each image is representative of 5 individual biological samples. (**G**) Immunostaining for tdT, β-catenin, GS, and E-CAD on liver sections collected at week 4 post-Tam. Scale bars, yellow 1 mm; white 100 μm. Each image is representative of 5 individual biological samples. (**H**) A schematic showing the experimental strategy. (**I**) Western blotting of β-catenin and GAPDH in sorted tdT+ cells. (**J**) Quantification of the relative expression of β-catenin protein. Data are the means ± SEM; n=3. **p<0.01 by Student's *t*-test. (**K**) qRT-PCR analysis of the relative expression of *Ctnnb1* in sorted tdT+ cells. Data are the means ± SEM; n=6. ****p<0.0001 by Student's *t*-test. (**L**) qRT-PCR analysis of the relative expression of *Glul*, *Axin2*, *Cyp1a2*, *Cyp2e1*, *Oat*, *Tcf7*, *Lect2*, *Tbx3*, *Slc1a2*, and *Rhbg* in sorted tdT+ cells. Data are the means ± SEM; n=6. ***p<0.001, ****p<0.0001 by Student's *t*-test.

The online version of this article includes the following source data and figure supplement(s) for figure 3:

**Source data 1.** Numerical data to generate *Figure 3*.

**Source data 2.** Original files for western blot analysis are displayed in *Figure 3I*.

**Source data 3.** PDF files containing original western blot for *Figure 3I*, indicating the relevant bands and treatments.

**Figure supplement 1.** Generation and characterization of the *Cyp2e1-DreER* mouse line.

if mCre/tdT could be expressed after *Stop* removal (*Figure 4C*), respectively, by *Rosa26-loxCre-tdT;Rosa26-tdT* (Strategy 1) and by injecting AAV8-Cre virus into *Rosa26-loxCre-tdT* mice (Strategy 2). Immunostaining for tdT on tissue sections revealed no tdT expression in Strategy 1, demonstrating minimal to negligible leakiness at the baseline. The robust tdT expression in the liver in Strategy 2 showed that mCre/tdT was allowed to express after *Stop* removal by AAV2/8-hTBG-Cre (*Figure 4D*). The FACS results confirmed that no leakiness was observed in *Rosa26-loxCre-tdT* (*Figure 4—figure supplement 1A*).

We then validate whether the new design can improve the labeling efficiency compared to the conventional approach by crossing endothelial cell-specific *Cdh5-CreER* with the conventional reporter *Rosa26-tdT* (Strategy 3) and the new reporter *Rosa26-loxCre-tdT* mice (Strategy 4) under the same protocol of Tam treatment (*Figure 4E*). Notably, immunostaining and FACS results showed that *Rosa26-loxCre-tdT* significantly limited the leakiness of adult *Cdh5-CreER* compared to *Rosa26-tdT* (*Figure 4—figure supplement 1B*). For the labeling efficiency comparison, immunostaining for tdT and VE-Cad on tissue sections revealed no difference in the percentage of VE-Cad+ endothelial cells expressing tdT between Strategies 3 and 4 (*Figure 4F–G*), suggesting that recombination of the *Rosa26-loxCre-tdT* allele by CreER was comparable to *Rosa26-tdT*, which is an easy-to-recombine reporter mouse line. Therefore, these data indicate that *Rosa26-loxCre-tdT* can be a convenient reporter, enabling efficient recombination to generate mCre/tdT in CreER-expressing cells without spontaneous leakiness.

### *Rosa26-loxCre-tdT* enables CreER to recombine *Rosa26-Confetti* efficiently

We next examined whether mCre released from loxCre could also display high recombination efficiency for the inert-to-recombine allele *Rosa26-Confetti*. We generated the triple knock-in *Cdh5-CreER;Rosa26-loxCre-tdT;Rosa26-Confetti* mouse line in which mCre was released for confetti recombination upon Tam treatment, and *Cdh5-CreER;Rosa26-Confetti* was used as the control (*Figure 5A*). Both mouse strains were treated with Tam at 8 weeks of age, and their tissue samples were collected for analysis at 1 week post-Tam (*Figure 5B*). As tdT expression is also detected in *Rosa26-loxCre-tdT* after the first recombination by CreER, we only compared the YFP and mCFP fluorescence signals of *Rosa26-Confetti* (*Figure 5A*). Whole-mount fluorescence imaging showed markedly more YFP and mCFP signals in the retina of the *Cdh5-CreER;Rosa26-loxCre-tdT;Rosa26-Confetti* mice, compared to *Cdh5-CreER;Rosa26-Confetti* mice (*Figure 5C*). Immunostaining on frozen sections revealed a significantly higher percentage of endothelial cells expressing YFP/mCFP in various vascularized organs of the *Cdh5-CreER;Rosa26-loxCre-tdT;Rosa26-Confetti* mice, compared to *Cdh5-CreER;Rosa26-Confetti* mice (*Figure 5D*). Notably, the results showed that tdT+ endothelial cells simultaneously expressed YFP/mCFP in *Cdh5-CreER;Rosa26-loxCre-tdT;Rosa26-Confetti* mice (*Figure 5D*). Thus, these data demonstrate that the *Rosa26-loxCre-tdT* line could be used as an adaptor to enhance CreER-mediated recombination with a simultaneous expression of tdT as an accurate indicator.

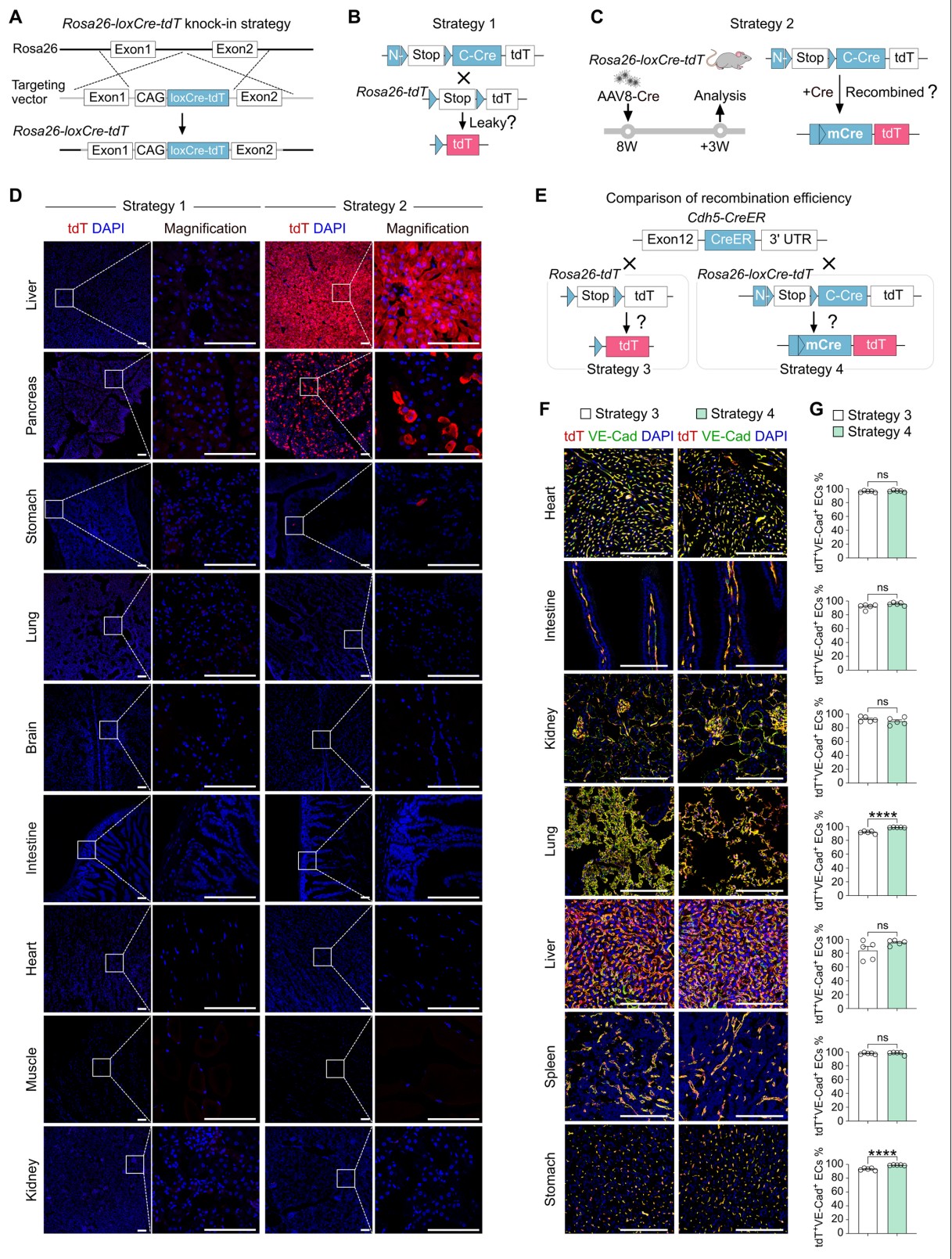

**Figure 4.** *Rosa26-loxCre-tdT* is efficiently recombined by CreER. (**A**) A schematic showing the knock-in strategy for the generation of the *Rosa26-loxCre-tdT* allele. In this line, tdT is used to denote the mCre expression after the removal of *Stop*. (**B**) A schematic showing experimental Strategy 1 to examine the leakiness of *Rosa26-loxCre-tdT* mice. (**C**) A schematic showing experimental Strategy 2 to test mCre/tdT expression by AAV2/8-hTBG-Cre. (**D**) Immunostaining for tdT on tissue sections shows tdT expression in the liver and pancreas when recombination was initiated by AAV2/8-hTBG-

*Figure 4 continued on next page*

Figure 4 continued

Cre. Scale bars, white 100 μm. Each image is representative of 5 individual biological samples. (E) A schematic showing the experimental Strategies 3 and 4 for comparing the CreER-mediated first recombination efficiency between *Cdh5-CreER;Rosa26-tdT* and *Cdh5-CreER;Rosa26-loxCre-tdT* mice. (F) Immunostaining for tdT and VE-Cad on liver sections collected on day 7 post-Tam. Scale bars, white 100 μm. Each image is representative of 5 individual biological samples. (G) Quantification of the percentage of VE-Cad$^+$ ECs expressing tdT. Data are the means ± SEM; n=5. ****p<0.0001 by Student's *t*-test.

The online version of this article includes the following source data and figure supplement(s) for figure 4:

Source data 1. Numerical data to generate *Figure 4G*.

Figure supplement 1. Characterization of the *Rosa26-loxCre-tdT* mouse line.

Figure supplement 1—source data 1. Numerical data to generate *Figure 4—figure supplement 1B*.

## *Rosa26-loxCre-tdT* adaptor ensures efficient recombination

To evaluate the specificity and efficiency of the loxCre strategy, we generated the *Alb-CreER;Rosa26-loxCre-tdT;Rosa26-Confetti* mouse line, in which *Alb-CreER* recombines *Rosa26-loxCre-tdT* allele in hepatocytes upon Tam treatment to release mCre/tdT, and mCre is expected to subsequently recombine the *Rosa26-Confetti* allele (*Figure 6A*, the right panel). As a control, we crossed the *Alb-CreER;Rosa26-Confetti* mouse with an alternative version of tdT reporter, *Rosa26-tdT2* mice (*Liu et al., 2020*), in which a long *Stop* cassette was inserted between two loxP sites to reduce the false-positive tracing results (*Figure 6A*, the left panel). Moreover, we also crossed *Alb-CreER;Rosa26-Confetti* with *iSuRe-Cre* (*loxP-NphiM-Stop-loxP-tdT-Int-Cre*) (*Fernández-Chacón et al., 2019*), in which the first recombination releases tdT and Cre expression, which is expected to subsequently recombine *Rosa26-Confetti* (*Figure 6A*, the middle panel). The *iSuRe-Cre* is a published and widely utilized inducible reporter-Cre mouse line, which has been reported to improve recombination activity in tdT$^+$ cells (*Fernández-Chacón et al., 2019*).

Since most hepatocytes are multinucleated and polyploid, a single hepatocyte would carry multiple copies of the *Rosa26-Confetti* alleles, yielding two or more fluorescence reporters. Therefore, we used tdT as the readout on *Alb-CreER*-mediated recombination of these three following reporters: *Rosa26-tdT2*, *iSuRe-Cre*, and *Rosa26-loxCre-tdT*. YFP/mCFP were used as readouts on *Rosa26-Confetti* (*Figure 6C*). We injected one dosage of Tam into *Alb-CreER;Rosa26-tdT2;Rosa26-Confetti*, *Alb-CreER;iSuRe-Cre;Rosa26-Confetti*, and *Alb-CreER;Rosa26-loxCre-tdT;Rosa26-Confetti* mice at 7 weeks of age and collected their livers for analysis at 1 week post-Tam (*Figure 6B*). The fluorescence images presented that the percentage of hepatocytes expressing tdT was highest in *Rosa26-loxCre-tdT* (91.76 ± 1.77%), followed by *Rosa26-tdT2* (35.24 ± 1.66%), and least in *iSuRe-Cre* (1.40 ± 0.13%) (*Figure 6D*), indicating the orders of recombination efficiency for these alleles were *Rosa26-loxCre-tdT>Rosa26-tdT2>iSuRe*-Cre. Furthermore, we compared the recombination efficiency on *Rosa26-Confetti* allele and found that the percentage of hepatocytes expressing YFP/mCFP was 1.08 ± 0.23% by *Alb-CreER;Rosa26-tdT2*, 19.42 ± 3.71% by *Alb-CreER;iSuRe-Cre*, and 100.00 ± 0.00% by *Alb-CreER;Rosa26-loxCre-tdT* (*Figure 6E*). Together, these findings collectively indicate that the mCre, when released from the *Rosa26-loxCre-tdT* construct, exhibits a significantly enhanced recombination efficiency. This improvement is notable when compared to both the conventional CreER and previously reported *iSuRe-Cre* (*Fernández-Chacón et al., 2019*).

## *Rosa26-loxCre-tdT* ensures gene deletion in tdT$^+$ cells

To investigate whether *Rosa26-loxCre-tdT* enables *Alb-CreER* to more efficiently delete genes, we crossed *Alb-CreER;Rosa26-loxCre-tdT* with *Ctnnb1*$^{flox/flox}$ to generate *Alb-CreER;Rosa26-loxCre-tdT;Ctnnb1*$^{flox/flox}$ mice for evaluating the efficiency of releasing mCre and also mCre-mediated *Ctnnb1* deletion in tdT$^+$ hepatocytes (*Figure 7A*). We also crossed *Alb-CreER;Rosa26-tdT2* with *Ctnnb1*$^{flox/flox}$ to generate *Alb-CreER;Rosa26-tdT2;Ctnnb1*$^{flox/flox}$ mice to evaluate recombination efficiency of both *Rosa26-tdT2* and *Ctnnb1*$^{flox/flox}$ alleles (*Figure 7A*). Additionally, *Alb-CreER;Rosa26-loxCre-tdT;Ctnnb1*$^{flox/+}$ mice were used as a control for heterozygous gene deletion (*Figure 7B*). First, the FACS and immunostaining data showed no leakiness in *Alb-CreER;Rosa26-loxCre-tdT* (*Figure 7—figure supplement 1A*). Based on the no leaky background, we treated *Alb-CreER;Rosa26-tdT2;Ctnnb1*$^{flox/flox}$ and *Alb-CreER;Rosa26-loxCre-tdT;Ctnnb1*$^{flox/+}$ mice with five dosages of Tam, and treated *Alb-CreER;Rosa26-loxCre-tdT;Ctnnb1*$^{flox/flox}$ mice only with one dosage of Tam, and collected hepatocytes

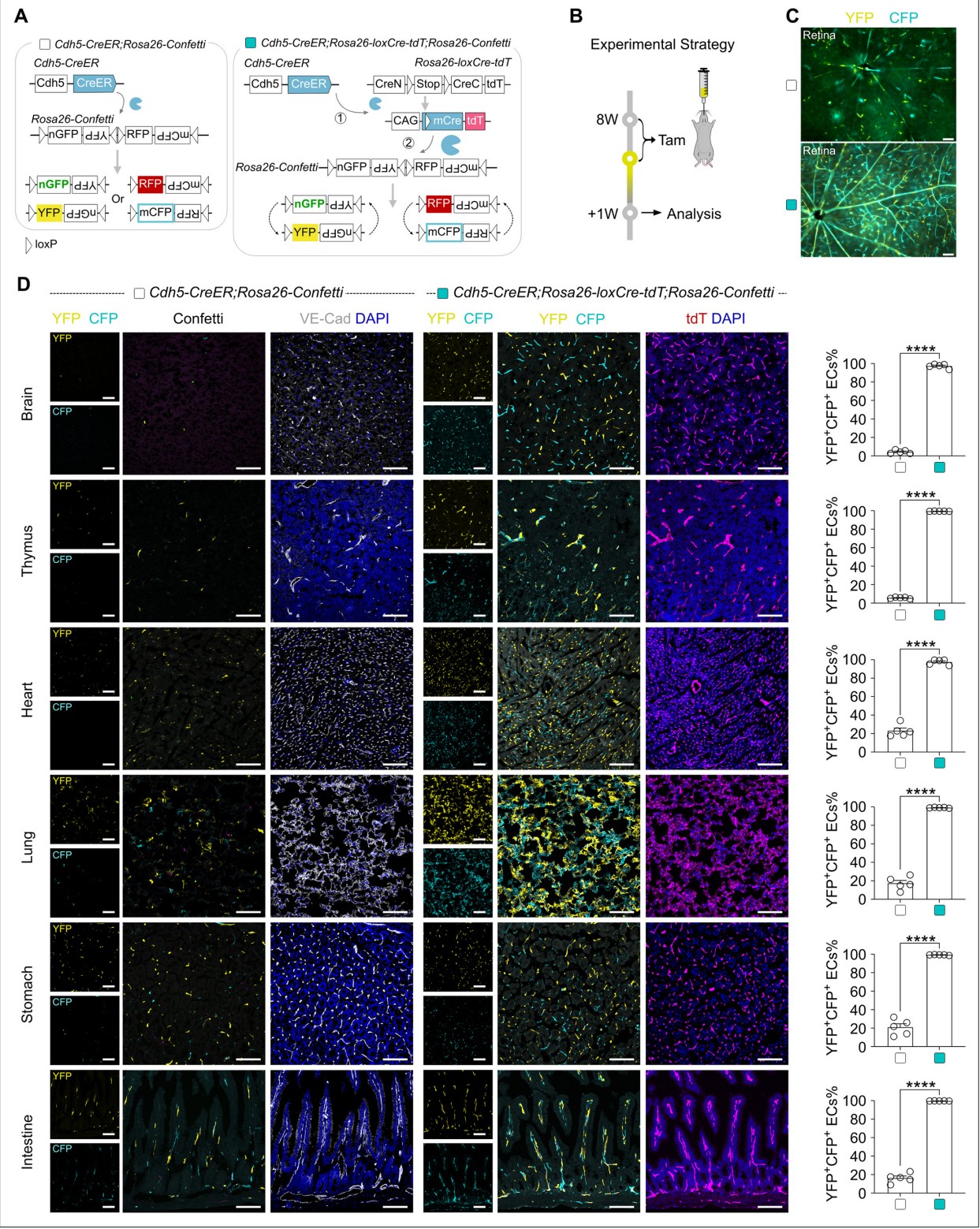

**Figure 5.** *Rosa26-loxCre-tdT* enables CreER to recombine *Rosa26-Confetti* efficiently. (**A**) Schematics showing the experimental design. In *Cdh5-CreER;Rosa26-loxCre-tdT;Rosa26-Confetti* mice, Tam-induced CreER-loxP recombination switches *Rosa26-loxCre-tdT* into *Rosa26-mCre-tdT* allele with simultaneous tdT labeling and expression of mCre, which subsequently targets *Rosa26-Confetti* (right panel). The conventional *Cdh5-CreER;Rosa26-Confetti* mice are used as controls. (**B**) A schematic showing the experimental strategy. (**C**) Whole-mount YFP and mCFP fluorescence images of retina collected from two mice groups. Scale bars, white 100 μm. Each image is representative of 5 individual biological samples. (**D**) Immunofluorescence

*Figure 5 continued on next page*

*Figure 5 continued*

images of YFP, mCFP, and VE-Cad on tissue sections show significantly more YFP+ and/or mCFP+ endothelial cells (ECs) in the *Cdh5-CreER;Rosa26-loxCre-tdT;Rosa26-Confetti* mice compared with those of *Cdh5-CreER;Rosa26-Confetti* mice (left panel). The right panel shows the quantification of ECs expressing YFP and/or mCFP. Data are the means ± SEM; n=5. ****p<0.0001 by Student's *t*-test. Scale bars, white 100 μm. Each image is representative of 5 individual biological samples.

The online version of this article includes the following source data for figure 5:

**Source data 1.** Numerical data to generate *Figure 5D*.

---

for analyzing *Ctnnb1* gene deletion at 4 weeks after Tam treatment (*Figure 7B*). Immunostaining data showed that β-catenin, GS, and E-CAD were significantly reduced in *Alb-CreER;Rosa26-loxCre-tdT;Ctnnb1 flox/flox* mice but were readily detectable in both *Alb-CreER;Rosa26-tdT2;Ctnnb1 flox/flox* and *Alb-CreER;Rosa26-loxCre-tdT;Ctnnb1 flox/+* mice (*Figure 7C*). Western blotting of β-catenin and qRT-PCR of *Ctnnb1* of isolated hepatocytes revealed the significant deletion of *Ctnnb1*

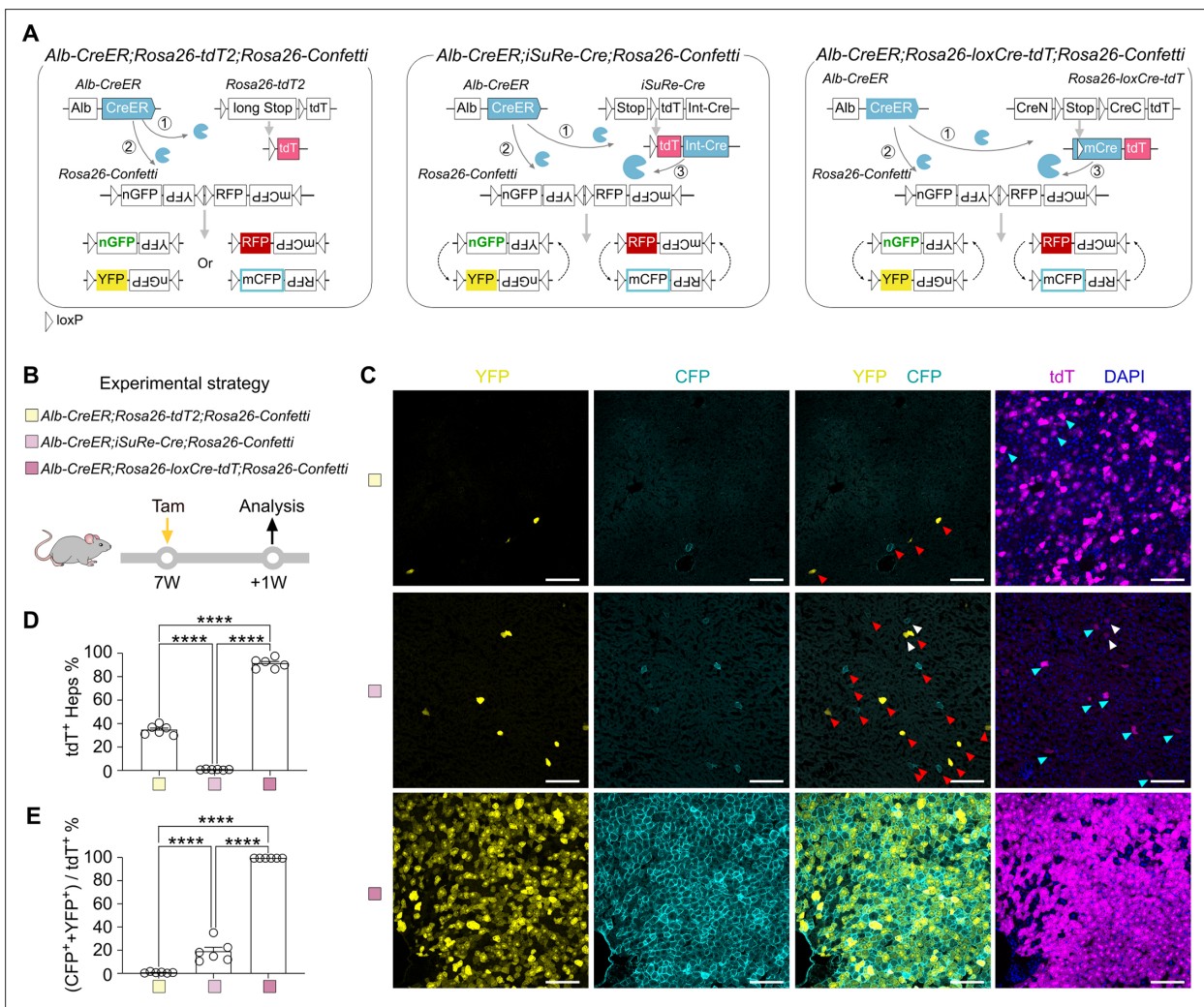

**Figure 6.** *Rosa26-loxCre-tdT* adaptor ensures efficient recombination. (**A**) The working strategies for *Alb-CreER;Rosa26-tdT2;Rosa26-Confetti*, *Alb-CreER;iSuRe-Cre;Rosa26-Confetti*, and *Alb-CreER;Rosa26-loxCre-tdT;Rosa26-Confetti*. (**B**) The experimental strategy. (**C**) Fluorescence images of YFP, mCFP, and tdT on the liver sections. The red arrows point out some tdT−YFP+ or tdT−mCFP+ hepatocytes. The white arrows point out some tdT+YFP+ or tdT+mCFP+ hepatocytes. The cyan arrows point out some tdT+YFP− or tdT+CFP− hepatocytes. (**D**) Quantification of the tdT+ hepatocytes. (**E**) Quantification of the percentage of tdT+ hepatocytes expressing YFP and/or mCFP. Data are means ± SEM; n=6. ****p<0.0001 by one-way ANOVA. Scale bars, white 100 μm. Each image is representative of 6 individual biological samples.

The online version of this article includes the following source data for figure 6:

**Source data 1.** Numerical data to generate *Figure 6*.

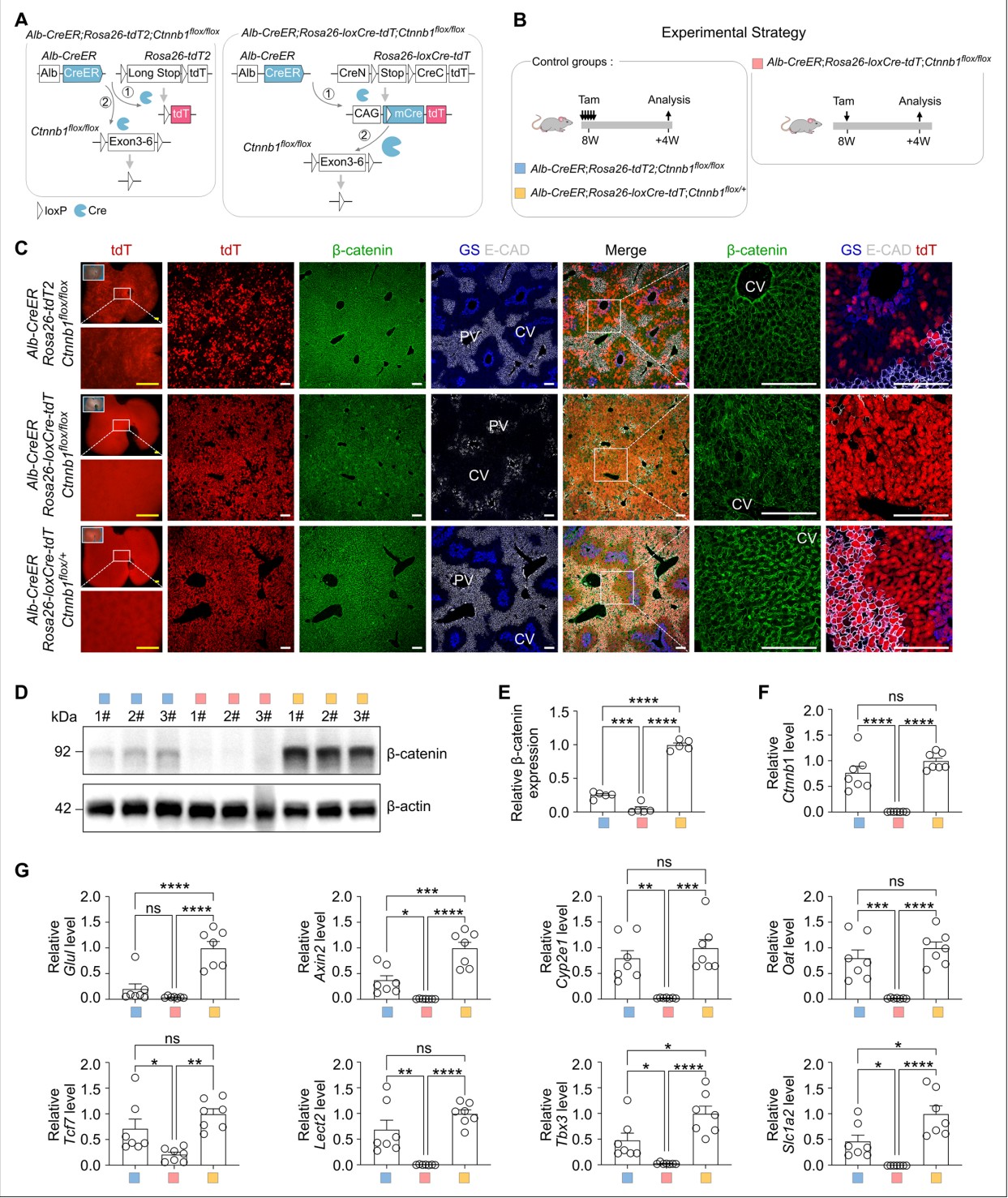

**Figure 7.** *Rosa26-loxCre-tdT* enables *Alb-CreER* to efficiently knock out genes in hepatocytes. (**A**) Schematics showing the experimental designs for *Ctnnb1* gene knockout using either *Alb-CreER;Rosa26-tdT2* or *Alb-CreER;Rosa26-loxCre-tdT* mice. (**B**) A schematic showing the experimental strategy. *Alb-CreER;Rosa26-tdT2;Ctnnb1^{flox/flox}* and *Alb-CreER;Rosa26-loxCre-tdT;Ctnnb1^{flox/+}* mice were injected with Tam for five times, while *Alb-CreER;Rosa26-loxCre-tdT;Ctnnb1^{flox/flox}* mice were injected with Tam once. (**C**) Immunostaining for tdT, β-catenin, GS, and E-CAD on liver sections. Scale bars, yellow 1 mm; white 100 μm. Each image is representative of 5 individual biological samples. (**D**) Western blotting of β-catenin and β-actin expression in hepatocytes, n=3. (**E**) Quantification of β-catenin expression. Data are the means ± SEM; n=5. (**F**) qRT-PCR analysis of the relative expression of *Ctnnb1* in the hepatocytes. (**G**) qRT-PCR analysis of the relative expression of *Glul, Axin2, Cyp2e1, Oat, Tcf7, Lect2, Tbx3,* and *Slc1a2* in hepatocytes. Data are the means ± SEM; n=7; ns, nonsignificant; *p<0.05, **p<0.01, ***p<0.001, ****p<0.0001 by one-way ANOVA.

*Figure 7 continued on next page*

*Figure 7 continued*

The online version of this article includes the following source data and figure supplement(s) for figure 7:

**Source data 1.** Numerical data to generate *Figure 7*.

**Source data 2.** Original files for western blot analysis are displayed in *Figure 7D*.

**Source data 3.** PDF files containing original western blot for *Figure 7D*, indicating the relevant bands and treatments.

**Figure supplement 1.** *Ctnnb1* was specifically knocked out in tdT⁺ hepatocytes in the *Alb-CreER;Rosa26-loxCre-tdT* mouse line.

**Figure supplement 1—source data 1.** Numerical data to generate *Figure 7—figure supplement 1*.

**Figure supplement 2.** Genotyping information for the newly established mouse lines in this study.

**Figure supplement 2—source data 1.** The original files for the agarose gel electrophoresis results are shown in *Figure 7—figure supplement 2*.

**Figure supplement 2—source data 2.** PDF files containing the agarose gel electrophoresis results of *Figure 7—figure supplement 2*, indicating the relevant bands and treatments.

---

in the *Alb-CreER;Rosa26-loxCre-tdT;Ctnnb1* ^flox/flox^ mice compared with those in control groups (*Figure 7D–F*). Additionally, Wnt downstream or related zonation-regulated genes, including *Glul*, *Axin2*, *Cyp2e1*, *Oat*, *Tcf7*, *Lect2*, *Tbx3*, and *Slc1a2*, were significantly reduced in hepatocytes derived from the *Alb-CreER;Rosa26-loxCre-tdT;Ctnnb1* ^flox/flox^ mice, compared with those from the *Alb-CreER;Rosa26-tdT2;Ctnnb1* ^flox/flox^ and *Alb-CreER;Rosa26-loxCre-tdT;Ctnnb1* ^flox/+^ mice, respectively (*Figure 7G*).

We next elucidate whether the significant superiority of *Alb-CreER;Rosa26-loxCre-tdT;Ctnnb1* ^flox/flox^ in the above data is attributed to low Tam dosage, which may reduce Tam toxicity and side effects. Initially, we treated *Alb-CreER;Rosa26-tdT;Ctnnb1* ^flox/flox^ mice with one dosage of Tam and subsequently sorted out tdT⁺ hepatocytes for qRT-PCR against *Ctnnb1* 4 weeks after Tam treatment. The results indicated that *Alb-CreER* with a single dose of Tam, without the assistance of *Rosa26-loxCre-tdT*, was insufficient to knock out *Ctnnb1* (*Figure 7—figure supplement 1B–D*). Next, we conducted an experiment involving a proportional gradient of the Tam dosages, in which we gave five dosages: one dosage, one-fifth of one dosage, one-twenty-fifth of one dosage, and one-hundred-twenty-fifth of one dosage of Tam to *Alb-CreER;Rosa26-loxCre-tdT;Ctnnb1* ^flox/flox^ mice, respectively. We analyzed samples 3 weeks after Tam's treatment. The FACS results showed that five groups of the proportional gradient of the Tam dosages labeled a gradient of tdT⁺ cells: 94.33 ± 0.76%, 90.14 ± 0.41%, 57.07 ± 0.51%, 27.30 ± 2.35%, and 1.27 ± 0.54% (*Figure 7—figure supplement 1E*), respectively. We sorted out tdT⁺ hepatocytes from each group for qRT-PCR against *Ctnnb1*. Our findings revealed that varying doses of Tam treatment primarily affect the target cell ratio but do not influence the high recombination efficiency of mCre at loxP sites within the *Ctnnb1* allele (*Figure 7—figure supplement 1E and F*). Additionally, the immunostaining analysis revealed a noticeable trend, in which the proportion of GS⁺ or E-CAD⁺ hepatocytes was decreased with the increase of target tdT⁺ hepatocytes (*Figure 7—figure supplement 1F and G*). This observation can be attributed to the significant knockout of *Ctnnb1* in tdT⁺ cells (*Figure 7—figure supplement 1E and F*).

Together, the combined results lead to the conclusion that *Rosa26-loxCre-tdT* significantly enhances the recombination efficiency of CreER, facilitating the targeting of genes that are challenging to manipulate by traditional CreER-loxP recombination. Furthermore, the tdT component in *Rosa26-loxCre-tdT* enables precise tracing of the modified cell lineage, which accurately investigates the roles of genes in various biological processes.

## Discussion

In this study, we demonstrated that the *rox-Stop-rox* insert placed in front of *Cre* does not effectively prevent the leakiness of Cre in the Rosa26 allele. To address this issue, we modified *Cre* by inserting the *RSR* cassette into its coding region, thereby dividing Cre into its N- and C-terminal segments. Our study showed that the developed roxCre not only eliminates leakage but also enables the release of mCre for targeted gene manipulation in specific cell subpopulations through the promotion of DreER. According to the construction of roxCre, we advanced to develop *Rosa26-loxCre-tdT*, which significantly enhances the efficiency of inducible Cre-mediated gene manipulation, particularly when using less effective CreER drivers for gene functional exploration.

---

Assuming successful gene deletion in the cell type of interest depends on efficient cell labeling by a lineage-specific reporter mediated by the same Cre is unrigorous, as recombination efficiency varies significantly among different mouse lines. Our data clearly showed the differences in recombination efficiency of different alleles (e.g. *Rosa26-tdT*, *Rosa26-tdT2*, and *Rosa26-Confetti*), even located in the same genomic locus (e.g. Rosa26), mediated by the same CreER under the same Tam treatment protocol (*Figures 6 and 7C* and *Figure 7—figure supplement 1D*). However, *Rosa26-loxCre-tdT* can be efficient for CreER-mediated recombination and release mCre in targeted cells, enabling efficient and accurate manipulation of cells of interest.

Compared to the previously reported transgenic mouse line, *iSuRe-Cre* (*Fernández-Chacón et al., 2019*), *Rosa26-loxCre-tdT* shows more stable and robust enhancement in the efficiency of CreER recombination (*Figures 6 and 7*), avoiding the toxic effects resulting from high doses of Tam. This advancement significantly improves the reliability of genetic manipulation and cell fate lineage tracing, making *Rosa26-loxCre-tdT* a valuable tool for researchers investigating cellular functions and behaviors. The research group of the iSuRe-Cre system recently reported a new system, named iSuRe-HadCre, which demonstrated to be effective in deleting floxed genes while avoiding the toxicity of constitutive Cre (*Garcia-Gonzalez et al., 2024*). Further studies can be carried out to compare the recombination enhancement between *Rosa26-loxCre-tdT* and *iSuRe-HadCre*. Given the high recombination efficiency of mCre, we envision that loxCre can be used for mosaic analysis, where multiple alleles in individual cells need to be efficiently recombined for evaluation of cell-autonomous gene function with genetic labeling.

Importantly, Cre toxicity represents a substantial concern. The toxicity may stem from the administration of Tam/4-OHT and the constitutive expression of Cre driven by specific promoters, including *Cd4*, *Ins2*, and *Myh6* (*Rashbrook et al., 2022*). Cre toxicity has been shown to impact diverse cellular processes, including DNA damage, cell proliferation, cell apoptosis, inflammation, metabolic signaling, and genetic dysfunction (*Brash et al., 2020*; *Rashbrook et al., 2022*). Although roxCre and loxCre can mitigate the toxicity associated with high-dose Tam, they cannot eliminate the Cre toxicity, as mCre is constitutively expressed following recombination. Considering the lack of Cre toxicity evaluation of the newly developed mouse lines in this study, Cre toxicity should be taken into account in future biological functional studies involving roxCre and loxCre systems. Furthermore, to optimize the robustness and rapidity of Cre expression while minimizing the toxic effects associated with constitutive Cre, further development of an iterated Cre removal system should be carried out for both loxCre and roxCre systems in the future.

Wnt-β-catenin plays an important role in liver development, zonation, homeostasis, and diseases (*Perugorria et al., 2019*). The administration of five doses of Tam in *Alb-CreER* mice did not impact downstream protein GS expression due to not knocking out *Ctnnb1* completely. However, a single dose of Tam could effectively induce a significant knockout of *Ctnnb1* through *Alb-CreER;Rosa26-loxCre-tdT* (*Figure 7*). This indicates that *Rosa26-loxCre-tdT* can assist CreER in addressing challenging recombined loxP sites. Notably, *Rosa26-loxCre-tdT* can release mCre by CreER-mediated activation and report tdT expression across all cells that express the recombinase. Therefore, this system does not produce a gradient of reporter fluorescence reflective of the promoter activity of CreER. Consequently, it's advisable not to use *Rosa26-loxCre-tdT* with a nonspecific and unstable CreER mouse line, which would amplify the CreER mouse line's weakness and result in unanticipated nonspecific cell targeting.

Furthermore, the strategy of roxCre further enhances the precision for cell fate mapping with genetic deletion simultaneously in Dre$^+$Cre$^+$ cells through intersectional genetic targeting. For example, we have showcased the strength of roxCre for genetic manipulation in peri-central hepatocytes, resulting in efficient gene knockout in tdT$^+$ hepatocytes specifically (*Figure 3*). Additional roxCre mouse lines driven by different promoters can serve as valuable tools for detailed investigations into specific cell subpopulations. By using roxCre, studies can achieve precise spatial and temporal control of gene expression, facilitating targeted analysis that contributes to the knowledge of developmental biology, disease mechanisms, and potential therapeutic interventions.

Previous studies have used Cre-loxP to simultaneously overexpress genes and fluorescence reporters in a cell for functional genetic mosaic (ifgMosaic) analysis (*Pontes-Quero et al., 2017*). However, ifgMosaic could not take advantage of the abundant resources from existing available floxed mouse lines for loss-of-function study. Mosaic mutant analysis with spatial and temporal control

of recombination (MASTR) utilized FlpoER for tissue-specific expression of GFPCre to delete floxed genes (*Lao et al., 2012*). Compared with FlpoER lines, CreER is a more broadly used recombinase that most laboratories use in gene deletion experiments. A recent study using dual recombinase-mediated cassette exchange (MADR) also permits stable labeling of mutant cells expressing transgenic elements from defined chromosomal loci (*Kim et al., 2019*). However, the use of viruses and electroporation, albeit convenient and rapid, is inefficient for specifically targeting any type of cells for in vivo studies. The powerful mosaic analysis with double markers (MADM) enables simultaneous lineage tracing of a pair of mutant and control sibling cells with distinct fluorescence reporters, allowing precise mosaic analysis of gene function in any cell (*Zong et al., 2005*; *Liu et al., 2011*). Nevertheless, the MADM cassette has to be combined with the mutant null allele, which is not readily available for synchronizing reporter expression and genetic modification in cells of interest. The ensured gene deletion with tdT reporter by *Rosa26-loxCre-tdT* could be coupled with the present CreER and loxP alleles for potential mosaic analysis. This could be achieved by adjusting Tam at a lower dosage so that the recombination efficiency to release mCre could be low in cells of interest for sparse tdT labeling. We believe that *Rosa26-loxCre-tdT* could be useful for robust, efficient, and temporal gene deletion at a population level and mosaic analysis at a single-cell level, with improvement in the future.

In conclusion, we developed two novel genetic tools, roxCre and loxCre, permitting efficient recombination to release mCre, which greatly enhanced the effectiveness of subsequent gene deletion in tdT⁺ cells, thus facilitating further detailed characterization of these cells both in situ and ex vivo. These tools are particularly beneficial when utilizing less effective DreER or CreER drivers to better elucidate the roles of specific genes in biological processes, facilitating more robust studies.

## Materials and methods

### Mice

Experiments using mice (*Mus musculus*) were carried out with the study protocols (SIBCB-S374-1702-001-C1) approved by the Institutional Animal Care and Use Committee of Center for Excellence in Molecular Cell Science (CEMCS), Shanghai Institute of Biochemistry and Cell Biology, Chinese Academy of Sciences. The *Rosa26-DreER* (*Li et al., 2018*), *Alb-CreER* (*He et al., 2017*), *Rosa26-GFP* (*Zhang et al., 2016a*), *Rosa26-tdT* (*Madisen et al., 2010*), *Rosa26-tdT2* (*Liu et al., 2020*), *Rosa26-RSR-tdT* (*Zhang et al., 2016b*), *Rosa26-Confetti* (*Snippert et al., 2010*), *iSuRe-Cre* (*Fernández-Chacón et al., 2019*), and *Ctnnb1-flox* (*Huelsken et al., 2001*) mouse lines were used as previously described. New knock-in mouse lines *Rosa26-RSR-Cre*, *Rosa26-RSR-Cre2*, *Rosa26-R-reverseCre-R*, *Cdh5-CreER*, *Alb-roxCre1-tdT*, *Alb-roxCre7-GFP*, *Cdh5-roxCre4-tdT*, *Cdh5-roxCre10-GFP*, *Cyp2e1-DreER*, and *Rosa26-loxCre-tdT* were generated by homologous recombination using CRISPR/Cas9 technology. These new mouse lines were generated by the Shanghai Model Organisms Center, Inc (SMOC). These mice were bred in a C57BL6/ICR mixed background. All mice were housed at the laboratory Animal Center of the Center for Excellence in Molecular Cell Science in a Specific Pathogen Free facility with individually ventilated cages. The room has controlled temperature (20–25°C), humidity (30–70%), and light (12 hr light-dark cycle). For the determination of the embryonic period of sampling, the day on which the vaginal plug was examined in female mice was recorded as E0.5. As the sex is not relevant to the topic of this study, male and female mice ranging in age from E13.5 to 12W (week) were allocated and mixed into the experimental groups in this study. No data in mouse experiments were excluded.

### Genomic PCR

Genomic DNA was prepared from the mouse toes or embryonic tails. Tissues were precipitated by centrifugation at maximum speed for 1 min at room temperature. After that, tissues were lysed by lysis buffer (100 mM Tris-HCl, 5 mM EDTA, 0.2% SDS, 200 mM NaCl, and 100 µg/mL Proteinase K) at 55°C overnight. About 750 µL pure ethanol was added to the lysis mixture and mixed thoroughly, followed by centrifugation at maximum speed for 5 min at room temperature to collect the DNA precipitation. Then the supernatant was discarded and the mixture was dried at 55°C for 1 hr. About 250 µL double-distilled H₂O was added to dissolve the DNA. The genomic PCR primer pairs were designed for the mutant alleles spanning both endogenous genomic fragments and insert fragments. The genetic constructs and genotyping of the new knock-in lines can be reviewed in *Figure 7—figure supplement 2*.

## In vitro screening of mCre for efficient recombination

To test if the remaining rox sequence after Dre-rox recombination would impact Cre activity, the pcDNA3.1 (*Paxinou et al., 2001*) (Invitrogen, V79020) was used to express 12 types of modified Cre (mCre). For testing the Cre activity, 500 ng pcDNA3.1-mCre plasmids, or pcDNA3.1 (negative control), or pHR-CMV-nlsCRE (positive control, Addgene, 12265) were mixed with 500 ng pCAG-loxP-Stop-loxP-ZsGreen (Addgene, 51269) plasmid. In vitro transfection experiment protocol is performed according to that described previously (*Jiang et al., 2021*). For cell culture medium, DMEM (Thermo Fisher, 11965092) was supplemented with 10% fetal bovine serum (Gibco, 10099141c) for preparing fresh complete culture medium. Poly-D-Lysine (Thermo Fisher, A3890401) precoated coverslips (Biosharp, BS-14-RC) or a 10 cm dish (Thermo Fisher, 150466) were used overnight to dry, and sterile water was used to flush them three times. HEK293A cells (ZQXZbio, ZQ0941) were identified by STR profiling and were negative for mycoplasma. A cryogenic vial of HEK293A was placed in a 37°C water bath, then the contents were thawed into a 50 mL conical tube (Corning, 430829) prefilled with 5 mL prewarmed fresh complete culture medium. After centrifuging at 125×*g* for 5 min, the supernatant was discarded, and the cells were resuspended in 10 mL complete medium. One-third of the cells were plated in a 10 cm coated dish and incubated in culture at 37°C. On the second day, the culture medium was discarded, and the cell layer was briefly rinsed with prewarmed PBS (Gibco, C10010500BT). After that, PBS was removed, and 1 mL 0.25% Trypsin solution (with EDTA, Gibco, 25200072) was added. The cells were incubated at 37°C for 3 min, and a 6 mL complete growth medium was added to aspirate cells by gently pipetting. The medium-containing cells were transferred to a 50 mL conical tube and centrifuged at 125×*g* for 5 min. The supernatant was discarded, and cells were resuspended in 5.2 mL complete culture medium. 24-Well plates were placed into coated coverslips, and 1 1 mL of complete culture medium was added. 200 µL of cells were added to each experimental well and incubated at 37°C for 9 hr to reach ~80% growth. The new prewarmed fresh complete culture medium replaced the old for 1 mL per well. Lipofectamine 3000 Transfection Reagent (Thermo Fisher, L3000015) was used for plasmid transfection. Samples were incubated in cultures at 37°C for 24 hr. Samples were washed with PBS once and mounted on slides with VECTASHIELD Antifade Mounting Medium with DAPI (Vector, H-1200). Images were acquired using an Olympus microscope (Olympus, BX53).

## Tamoxifen injection and sample collection

For induction of CreER or DreER, tamoxifen (Sigma, T5648) was dissolved in corn oil and administered to mice by oral gavage. *Cyp2e1-DreER;Alb-roxCre-tdT;Ctnnb1* *flox/+* and *Cyp2e1-DreER;Alb-roxCre-tdT;Ctnnb1* *flox/flox* mice were treated with 0.05 mg Tam per gram mouse body weight (0.05 mg/g) for one dose. For comparing tdT[+] cells, *Cyp2e1-DreER;Rosa26-RSR-tdT* and *Cyp2e1-Alb-roxCre-tdT* mice were treated with 0.05 mg/g Tam. *Rosa26-DreER;Alb-roxCre1-tdT;Rosa26-Confetti*, *Rosa26-DreER;Cdh5-roxCre4-tdT;Rosa26-Confetti*, *Rosa26-DreER;Alb-roxCre7-GFP;Rosa26-Confetti*, *Rosa26-DreER;Cdh5-roxCre10-GFP;Rosa26-Confetti*, *Rosa26-DreER;Alb-roxCre1-tdT;Rosa26-GFP*, *Rosa26-DreER;Alb-roxCre7-GFP;Rosa26-tdT*, *Alb-CreER;Rosa26-Confetti*, *Cdh5-CreER;Rosa26-Confetti*, *Alb-CreER;Rosa26-tdT2;Rosa26-Confetti*, *Alb-CreER;iSuRe-Cre;Rosa26-Confetti*, *Alb-CreER;Rosa26-loxCre-tdT;Rosa26-Confetti*, *Alb-CreER;Rosa26-loxCre-tdT;Ctnnb1* *flox/flox*, and *Alb-CreER;Rosa26-tdT;Ctnnb1* *flox/flox* mice were treated with 0.2 mg/g Tam for one dose. *Cdh5-CreER;Rosa26-tdT*, *Cdh5-CreER;Rosa26-loxCre-tdT*, *Cdh5-CreER;Rosa26-Confetti*, *Cdh5-CreER;Rosa26-loxCre-tdT;Rosa26-Confetti*, *Alb-CreER;Rosa26-tdT2;Ctnnb1* *flox/flox*, and *Alb-CreER;Rosa26-loxCre-tdT;Ctnnb1* *flox/+* mice were treated with 0.2 mg/g Tam one dose per day for 5 days. *Rosa26-RSR-Cre2;Rosa26-GFP* and *Rosa26-R-reverseCre-R;Rosa26-tdT* mice were sacrificed at 8 weeks of age for analysis. For the proportional gradient of the Tam dosages, five dosage groups were given 0.2 mg/g Tam for constitutive 5 days, one-fifth of one dosage group was given 0.04 mg/g Tam for one dose, one-twenty-fifth of one dosage group was given 0.008 mg/g Tam for one dose, and one-hundred-twenty-fifth of one dosage group was given 0.0016 mg/g Tam for one dose. Mice, both males and females, aged 8–12 weeks, were used in experiments with similarly aged mice in both the control and experimental groups.

## Whole-mount imaging and sectioning

The tissue samples were fixed with 4% paraformaldehyde (PFA) (Sigma, P6148-500g, wt/vol in PBS) for 1 or 2 hr (stomach, intestine, colon) at 4°C, followed by washing with PBS three times. The fixed tissues were placed in an agarose-filled Petri dish for bright-field and fluorescence imaging by a Zeiss stereoscopic microscope (AxioZoom V16). For cryo-sections, tissues were sectioned to slides of 10 μm thickness after dehydration by 30% sucrose (Sinopharm, H-10021463, wt/vol in PBS) overnight and pre-embedding with OCT (Scigen, 4586) at 4°C for 1 hr.

## Immunostaining

Immunostaining was performed as previously described (*Pu et al., 2023*). 0.2% PBST was prepared by dissolving 0.2% (vol/vol) Triton X-100 (Sigma, T9284) in PBS. Tissue sections were blocked with 2.5% normal donkey serum and 0.1% 4'6-diamidino-2-phenylindole (DAPI, Invitrogen, D21490) dissolved in 0.2% PBST for 30 min after washing with PBS three times. The tissue sections were incubated with the primary antibody diluted in 0.2% PBST at 4°C overnight. The next day, sections were incubated with secondary antibodies diluted in 0.2% PBST at room temperature for 30 min, followed by PBS washing three times. The slides were washed three times with PBS. The slides were mounted with a mounting medium (Vector Lab). For weak signals, the endogenous peroxidase activity was quenched before blocking. Horseradish peroxidase or biotin-conjugated secondary antibodies, and a Tyramide signal amplification kit (PerkinElmer), were used after incubation with the primary antibodies. For primary antibodies of murine origin, mouse immunoglobulins were blocked with an anti-mouse Fab antibody (Jackson, 715-007-003, 1:100). The included primary antibodies are listed as follows: tdT (Rockland, 600-401-379, 1:500; or Rockland, 200-101-379, 1:500), GFP (Invitrogen, A11122, 1:500), GFP (Rockland, 600-101-215M, 1:500), GFP (Nacalai, 04404–84; 1:500), GS (Abcam, ab49873, 1:1000), β-catenin (BD Pharmingen, 610153, 1:200), E-CAD (R&D, AF748, 1:500), Ep-CAM (Abcam, ab92382, 1:500), VE-Cad (R&D, AF1002, 1:100), CK19 (AbboMax, 602–670, 1:500), and HNF4α (Cell Signaling Technology, 3113, 1:1000). The corresponding secondary antibodies (JIR or Abcam) were diluted according to the instructions. Images were captured using a Nikon confocal (Nikon A1 FLIM) or an Olympus confocal (FV3000), and analyzed using ImageJ2 (version 2.9.0/1.54f) and Photoline (version 23.02). We collected five random fields from each liver section for quantification. Mutant and control sections were processed simultaneously to avoid potential batch differences during staining. Imaging of all immunostained slides was performed under the same exposure and contrast conditions using the same confocal microscope.

## Imaging of Confetti fluorescence reporters

All the fluorescence images of *Rosa26-Confetti* were taken by a Leica confocal (Leica SP8 will). All the sections were collected freshly and blocked with 0.2% PBST containing DAPI for 30 min after washing with PBS three times. The parameters of the excitation light and emission light for all channels are set as follows: DAPI (ex. 405 nm, em. 415–450 nm), CFP (ex. 458 nm, em. 463–481 nm), GFP (ex. 488 nm, em. 495–508 nm), YFP (ex. 514 nm, em. 520–535 nm), tdT (ex. 546 nm, em. 555–590 nm), markers for antibody staining (ex. 647 nm, em. 657–700 nm). Captured images were analyzed using ImageJ2 (version 2.9.0/1.54f) and Photoline (version 23.02).

## Hepatocyte dissociation

Mouse primary hepatocytes were isolated by the two-step collagenase perfusion method, which was modified from a previous protocol (*Charni-Natan and Goldstein, 2020*). Briefly, mice were anesthetized, and the liver was exposed through an incision in the lower abdomen. The inferior vena cava and portal vein were also exposed. A needle was inserted into the inferior vena cava and secured with a hemostatic clamp. The portal vein was cut immediately when the mouse liver was perfused with a perfusion medium buffer (containing 0.5 mM EGTA) for 5 min using a peristaltic pump. Then, the liver was perfused with medium containing collagenase type I (150 U/mL; Gibco, 17100-017) for 2–5 min to adequately digest the liver. After the gallbladder was removed, the liver was dissected with cold DMEM to free the hepatic cells. Then the cell suspension was passed through a 70 μm cell strainer (BD Biosciences, 352350) and centrifuged at 50×*g* for 3 min at 4°C. The supernatant was removed, and cells were resuspended in Percoll (GE Healthcare, 17-0891-01)/DMEM/10× PBS (Sangon, B548117, diluted in 1:1) (1:1:0.1) mixture and centrifuged at 300×*g* for 5 min at 4°C. After the supernatant was

removed, cells were dissected with cold DMEM and centrifuged at 50×*g* for 3 min at 4°C. Purified hepatocytes were collected for FACS analysis, qRT-PCR, or western blot analysis.

## Flow cytometric analysis and isolation of tdT⁺ hepatocytes

Cells were centrifuged at 50×*g* for 3 min at 4°C, then resuspended in the relevant solution. The suspended solution is 0.1% DNase I (Worthington, LS002139, diluted by DMEM) mixed with 0.1% DAPI. tdT⁺ hepatocytes were sorted by the FACS Aria SORP machine and the Sony MA900. Hepatocytes were collected by DMEM and then centrifuged at 50×*g* for 3 min at 4°C.

## Hepatic non-parenchymal cell dissociation

The liver was diced into 1 mm pieces and then transferred into 10 mL digestion buffer (0.05 g/100 mL collagenase type IV [Worthington, LS004188], 2 mg/mL collagenase type I, 5 U/mL Dispase [Corning, 354235], and 1% DNase I in PBS). The sample was incubated in a 37°C incubator shaker at 220 rotations per minute for 30 min, with mixing performed three times during the incubation. Following this, 0.5 mL FBS was added to the digestion mixture. The resulting cell suspension was passed through a 40 μm cell strainer (BD Biosciences, 352340) and transferred to a new tube containing 30 mL cold PBS after being centrifuged at 50×*g* for 3 min at 4°C. The supernatant was discarded after 700×*g* for 5 min at 4°C. To lyse erythrocytes, RBC lysis buffer was added, and the mixture was kept at room temperature for 5 min. Lysis was halted by adding 20 mL of cold PBS, followed by a second centrifugation at 700×*g* for 5 min at 4°C. The supernatant was discarded, and the pellet was resuspended in the suspension solution.

## Total RNA extraction and qRT-PCR

Total RNA was extracted from hepatocytes isolated from the indicated mice as previously described (*Pu et al., 2016*). Cells were lysed with TRIzol (Invitrogen, 15596018), and total RNA was extracted according to the manufacturer's instructions. For each sample, 1 μg of total RNA was reverse-transcribed into cDNA using the Prime Script RT kit (Takara, RR047A). The SYBR Green qPCR master mix (Thermo Fisher Scientific, 4367659) was used, and quantitative RT-PCR was performed on the QuantStudio 6 Real-Time PCR System (Thermo Fisher Scientific). GAPDH was used as the internal control. Sequences of all primers would be provided upon request.

## Western blot

All samples were lysed in RIPA lysis buffer (Beyotime, P0013B) containing protease inhibitors (Roche, 11836153001) for 30 min on ice, and then centrifuged at 15,000×*g* for 15 min to collect the supernatant. All samples were diluted to 30 μL for testing protein concentration using the Pierce BCA Protein Assay Kit (Thermo Scientific, 23227). The remaining samples were mixed with 5× loading buffer (Beyotime, p0015L) and boiled at 100°C for 5 min. 40 μg samples were added to each well in precast gradient gels (Beyotime, P0469M) with 1× running buffer (Epizyme, PS105S, diluted 1×). After running, samples were transferred onto Immobilon PVDF membranes (Millipore, IPVH00010). After blocking in the blocking buffer (Epizyme, PS108P), the membranes were incubated with primary antibodies (diluted by primary antibody dilution buffer [Epzyme, PS114]) overnight at 4°C, then washed three times and incubated with HRP-conjugated secondary antibodies (diluted by 1× TBST [Epzyme, PS103S, should be diluted into 1×]). Samples were incubated with chemiluminescent HRP substrate (Millipore, WBKLS0500), and related signals were detected by MiniChemi 610 Plus (Biogp). The following antibodies were used: β-catenin (BD Biosciences, 610153, 1:5000), GAPDH (Proteintech, 60004-1-IG, 1:2000), β-actin (Epzyme, LF202, 1:1000), HRP-donkey-a-mouse (JIR, 715-035-150, 1:5000), and HRP-goat a rabbit IgG (JIR, 111-035-047, 1:5000).

## AAV injection

The AAVs used in this study were purchased from Taitool Biotechnologies company (Shanghai, China). Briefly, AAVs were produced with cis-plasmids containing the full TBG promoter, which is specifically active in hepatocytes, and Cre expression is under the control of the TBG promoter. A replication-incompetent AAV2/8-hTBG-Cre virus (AAV8-hTBG-Cre, S0657-8-H5) was packaged and purified before application to mice. Mice were injected intraperitoneally at 1×10¹¹ genome copies of the virus

per mouse. The above newly generated virus, as well as targeting plasmids, will be provided upon request.

## Statistical analysis

Each pot in every graph represented one individual mouse. Quantification of each mouse from the fluorescence images was performed by averaging the counts from five 10× fields across different sections. Statistical analyses were performed using GraphPad Prism (version 9.5.1). All the continuous variables were expressed as means ± standard error of the mean (SEM). One-way ANOVA was used to detect statistical significance between three experimental groups. The statistical difference between the two experimental groups was determined using the unpaired Student's $t$-test. Probabilities (p)<0.05 were considered statistically significant. The equation, Log(agonist) vs. response – variable slope (four parameters) in Prism, was adapted for the dose-response curve for the proportional gradient of the Tam dosages experiment analysis, with a confidence interval of 95%.

## Acknowledgements

This study was supported by the National Key Research & Development Program of China (2024YFA1803302, 2023YFA1800700, 2022YFA1104200, 2023YFA1801300, 2020YFA0803202), National Natural Science Foundation of China (82088101, 32370897, 32100648, 32370783, 32100592), CAS Project for Young Scientists in Basic Research (YSBR-012), Shanghai Pilot Program for Basic Research-CAS, Shanghai Branch (JCYJ-SHFY-2021-0), Research Funds of Hangzhou Institute for Advanced Study (2022ZZ01015, B04006C01600515), Shanghai Municipal Science and Technology Major Project, and the New Cornerstone Science Foundation through the New Cornerstone Investigator Program and the XPLORER PRIZE, and CAS-Croucher Funding Scheme for Joint Laboratories.

## Additional information

### Funding

| Funder | Grant reference number | Author |
| --- | --- | --- |
| National Key Research & Development Program of China | 2020YFA0803202 | Bin Zhou |
| National Natural Science Foundation of China | 32100592 | Bin Zhou |
| CAS Project for Young Scientists in Basic Research | YSBR-012 | Bin Zhou |
| Shanghai Pilot Program for Basic Research-CAS, Shanghai Branch | JCYJ-SHFY-2021-0 | Bin Zhou |
| Research Funds of Hangzhou Institute for Advanced Study | B04006C01600515 | Bin Zhou |
| Shanghai Municipal Science and Technology Major Project | | Bin Zhou |
| New Cornerstone Science Foundation | New Cornerstone Investigator Program | Bin Zhou |
| XPLORER PRIZE | | Bin Zhou |
| CAS-Croucher Funding Scheme for Joint Laboratories | | Bin Zhou |
| National Key Research & Development Program of China | 2023YFA1801300 | Bin Zhou |

| Funder | Grant reference number | Author |
|---|---|---|
| National Key Research & Development Program of China | 2022YFA1104200 | Bin Zhou |
| National Key Research & Development Program of China | 2023YFA1800700 | Bin Zhou |
| National Key Research & Development Program of China | 2024YFA1803302 | Bin Zhou |
| National Natural Science Foundation of China | 32370783 | Bin Zhou |
| National Natural Science Foundation of China | 32100648 | Bin Zhou |
| National Natural Science Foundation of China | 32370897 | Bin Zhou |
| National Natural Science Foundation of China | 82088101 | Bin Zhou |
| Research Funds of Hangzhou Institute for Advanced Study | 2022ZZ01015 | Bin Zhou |

The funders had no role in study design, data collection and interpretation, or the decision to submit the work for publication.

## Author contributions

Mengyang Shi, Resources, Data curation, Software, Formal analysis, Validation, Investigation, Visualization, Methodology, Writing – review and editing; Jie Li, Data curation; Xiuxiu Liu, Resources, Bred the mice and performed experiments; Kuo Liu, Resources, Bred the mice and performed experiments; Lingjuan He, Resources, Data curation, Bred the mice and performed experiments; Wenjuan Pu, Resources, Bred the mice and performed experiments; Wendong Weng, Resources, Bred the mice and performed experiments; Shaohua Zhang, Resources, Data curation, Bred the mice and performed experiments; Huan Zhao, Resources, Bred the mice and performed experiments; Kathy Lui, Writing – review and editing, Provided intellectual input; Bin Zhou, Conceptualization, Resources, Data curation, Supervision, Funding acquisition, Validation, Methodology, Writing – original draft, Project administration, Writing – review and editing

## Author ORCIDs

Mengyang Shi https://orcid.org/0009-0005-7805-5436
Jie Li https://orcid.org/0009-0002-5550-5148
Kuo Liu https://orcid.org/0000-0003-0757-7833
Lingjuan He https://orcid.org/0000-0003-2747-5148
Wendong Weng https://orcid.org/0000-0003-3730-7965
Shaohua Zhang https://orcid.org/0000-0002-7300-4091
Huan Zhao https://orcid.org/0000-0001-6319-949X
Kathy Lui https://orcid.org/0000-0002-1616-3643
Bin Zhou https://orcid.org/0000-0001-5278-5522

## Ethics

This study was performed in strict accordance with the recommendations in the Institutional Animal Care and Use Committee of Center for Excellence in Molecular Cell Science (CEMCS), Shanghai Institute of Biochemistry and Cell Biology, Chinese Academy of Sciences. All of the animals were handled according to the Institutional Animal Care and Use Committee protocols (SIBCB-S374-1702-001-C1) of the Center for Excellence in Molecular Cell Science. Every effort was made to minimize suffering.

Reviewer #1 (Public review): https://doi.org/10.7554/eLife.97717.4.sa1
Reviewer #2 (Public review): https://doi.org/10.7554/eLife.97717.4.sa2

Reviewer #3 (Public review): https://doi.org/10.7554/eLife.97717.4.sa3
Author response https://doi.org/10.7554/eLife.97717.4.sa4

## Additional files

### Supplementary files

Supplementary file 1. Mouse genotypes. The subheadings in this document correspond to individual figures. The leftmost column indicates the lettered labels within each figure. The middle column provides descriptive annotations for the experimental groups. The right column specifies the detailed mouse genotypes for each respective group.

Supplementary file 2. Oligos for *M. musculus* genes. The leftmost column indicates the names of genes. The middle column provides related sequences. The right column describes the additional information on primers.

MDAR checklist

### Data availability

All data generated or analysed during this study are included in the manuscript. No further datasets and code were generated from this article.

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

# Appendix 1

## Appendix 1—key resources table

| Reagent type (species) or resource | Designation | Source or reference | Identifiers | Additional information |
|---|---|---|---|---|
| Strain, strain background (*Mus musculus*) | *Rosa26-DreER* | *Li et al., 2018* | 10.1161/CIRCULATIONAHA.118.034250 | Generated in a previous study |
| Strain, strain background (*Mus musculus*) | *Alb-CreER* | *He et al., 2017* | 10.1038/nm.4437 | Generated in a previous study |
| Strain, strain background (*Mus musculus*) | *Rosa26-GFP* | *Zhang et al., 2016a* | 10.1161/CIRCRESAHA.116.308749 | Generated in a previous study |
| strain, strain background (*Mus musculus*) | *Rosa26-tdT* | *Madisen et al., 2010* | 10.1038/nn.2467 | Generated in a previous study |
| Strain, strain background (*Mus musculus*) | *Rosa26-tdT2* | *Liu et al., 2020* | 10.1074/jbc.RA119.011349 | Generated in a previous study |
| Strain, strain background (*Mus musculus*) | *Rosa26-RSR-tdT* | *Zhang et al., 2016a* | 10.1161/CIRCRESAHA.115.307202 | Generated in a previous study |
| Strain, strain background (*Mus musculus*) | *Rosa26-Confetti* | *Snippert et al., 2010* | 10.1016/j.cell.2010.09.016 | Generated in a previous study |
| Strain, strain background (*Mus musculus*) | *iSuRe-Cre* | *Fernández-Chacón et al., 2019* | 10.1038/s41467-019-10239-4 | Generated in a previous study |
| Strain, strain background (*Mus musculus*) | *Ctnnb1-flox* | *Huelsken et al., 2001* | doi: 10.1016/s0092-8674(01)00336-1 | Generated in a previous study |
| Strain, strain background (*Mus musculus*) | *Rosa26-RSR-Cre* | This paper | This paper | Generated in this study and available from the corresponding author upon request |
| Strain, strain background (*Mus musculus*) | *Rosa26-RSR-Cre2* | This paper | This paper | Generated in this study and available from the corresponding author upon request |
| Strain, strain background (*Mus musculus*) | *Rosa26-R-reverseCre-R* | This paper | This paper | Generated in this study and available from the corresponding author upon request |
| Strain, strain background (*Mus musculus*) | *Cdh5-CreER* | This paper | This paper | Generated in this study and available from the corresponding author upon request |
| Strain, strain background (*Mus musculus*) | *Alb-roxCre1-tdT* | This paper | This paper | Generated in this study and available from the corresponding author upon request |
| Strain, strain background (*Mus musculus*) | *Alb-roxCre7-GFP* | This paper | This paper | Generated in this study and available from the corresponding author upon request |
| Strain, strain background (*Mus musculus*) | *Cdh5-roxCre4-tdT* | This paper | This paper | Generated in this study and available from the corresponding author upon request |

*Appendix 1 Continued on next page*

*Appendix 1 Continued*

| Reagent type (species) or resource | Designation | Source or reference | Identifiers | Additional information |
|---|---|---|---|---|
| Strain, strain background (*Mus musculus*) | *Cdh5-rox10-GFP* | This paper | This paper | Generated in this study and available from the corresponding author upon request |
| Strain, strain background (*Mus musculus*) | *Cyp2e1-DreER* | This paper | This paper | Generated in this study and available from the corresponding author upon request |
| Strain, strain background (*Mus musculus*) | *Rosa26-loxCre-tdT* | This paper | This paper | Generated in this study and available from the corresponding author upon request |
| Transfected construct (*Mus musculus*) | AAV2/8-TBG-Cre | Taitool: Cat# S0657-8-H5 | | AAV2/8 vector expressing Cre under the TBG promoter |
| Recombinant DNA reagent | pcDNA3.1 (plasmid) | Invitrogen: Cat# V79020 | | Parental plasmid from which mutants were generated |
| Recombinant DNA reagent | pCAG-loxp-stop-loxp-ZsGreen (plasmid) | Addgene: Cat# 51269 | | Reporter plasmid used to monitor Cre activity |
| Recombinant DNA reagent | pHR-CMV-nlsCRE (plasmid) | Invitrogen: Cat# 12265 | | Vector used to express Cre recombinase |
| Recombinant DNA reagent | pcDNA3.1-mCre1 (plasmid) | This paper | | Vector used to express mCre1 recombinase |
| Recombinant DNA reagent | pcDNA3.1-mCre2 (plasmid) | This paper | | Vector used to express mCre2 recombinase |
| Recombinant DNA reagent | pcDNA3.1-mCre3 (plasmid) | This paper | | Vector used to express mCre3 recombinase |
| Recombinant DNA reagent | pcDNA3.1-mCre4 (plasmid) | This paper | | Vector used to express mCre4 recombinase |
| Recombinant DNA reagent | pcDNA3.1-mCre5 (plasmid) | This paper | | Vector used to express mCre5 recombinase |
| Recombinant DNA reagent | pcDNA3.1-mCre6 (plasmid) | This paper | | Vector used to express mCre6 recombinase |
| Recombinant DNA reagent | pcDNA3.1-mCre7 (plasmid) | This paper | | Vector used to express mCre7 recombinase |
| Recombinant DNA reagent | pcDNA3.1-mCre8 (plasmid) | This paper | | Vector used to express mCre8 recombinase |
| Recombinant DNA reagent | pcDNA3.1-mCre9 (plasmid) | This paper | | Vector used to express mCre9 recombinase |
| Recombinant DNA reagent | pcDNA3.1-mCre10 (plasmid) | This paper | | Vector used to express mCre10 recombinase |
| Recombinant DNA reagent | pcDNA3.1-mCre11 (plasmid) | This paper | | Vector used to express mCre11 recombinase |
| Recombinant DNA reagent | pcDNA3.1-mCre12 (plasmid) | This paper | | Vector used to express mCre12 recombinase |
| Cell line (*Homo sapiens*) | HEK293A | ZQXZbio: Cat# ZQ0941 | RRID:CVCL_6910 | |
| Antibody | Donkey Polyclonal anti-mouse Fab antibody | Jackson: Cat# 715-007-003 | RRID:AB_2307338 | (20 µg/mL). Used for minimizing primary antibody cross-linking and avoiding Fc-mediated artifacts |
| Antibody | Rabbit Polyclonal anti-RFP | Rockland: Cat# 600-401-379 | RRID:AB_2209751 | IF (1:1000) |
| Antibody | Goat Polyclonal anti-RFP | Rockland: Cat# 200-101-379 | RRID:AB_2744552 | IF (1:1000) |

*Appendix 1 Continued on next page*

*Appendix 1 Continued*

| Reagent type (species) or resource | Designation | Source or reference | Identifiers | Additional information |
|---|---|---|---|---|
| Antibody | Rabbit Polyclonal anti-GFP | Invitrogen: Cat# A11122 | RRID:AB_221569 | IF (1:500) |
| Antibody | Goat Polyclonal anti-GFP | Rockland: Cat# 600-101-215M | RRID:AB_2612804 | IF (1:500) |
| Antibody | Rat monoclonal anti-GFP | Nacalai: Cat# 04404-84 | RRID:AB_2313654 | IF (1:500) |
| Antibody | Rabbit Polyclonal anti-Glutamine Synthetase | Abcam: Cat# ab49873 | RRID:AB_880241 | IF (1:10,000) |
| Antibody | Mouse monoclonal anti-β-catenin | BD Pharmingen: Cat# 610153 | RRID:AB_397555 | IF (1:200) WB (1:5000) |
| Antibody | Goat Polyclonal anti-E-cadherin | R&D: Cat# AF748 | RRID:AB_355568 | IF (1:500) |
| Antibody | Rat monoclonal anti-EpCAM | Abcam: Cat# ab92382 | RRID:AB_2049615 | IF (1:2000) |
| Antibody | Goat Polyclonal anti-VE-cadherin | R&D: Cat# AF1002 | RRID:AB_2077789 | IF (1:100) |
| Antibody | Rabbit Polyclonal anti-CK19 | AbboMax: Cat# 602-670 | RRID:AB_3720929 | IF (1:500) |
| Antibody | Rabbit monoclonal HNF4α (C11F12) Rabbit mA | Cell Signaling Technology: Cat# 3113 | RRID:AB_2295208 | IF (1:1000) |
| Antibody | Mouse monoclonal anti-GAPDH | Proteintech: Cat# 60004-1-IG | RRID:AB_2107436 | WB (1:5000) |
| Antibody | Rabbit Polyclonal anti-β-actin | Epizyme: Cat# LF202 | RRID:AB_3094632 | WB (1:5000) |
| Antibody | Donkey Polyclonal HRP-conjugated Donkey anti-mouse IgG | JIR: Cat# 715-035-150 | RRID:AB_2340770 | WB (1:5000) |
| Antibody | Goat Polyclonal HRP-conjugated Goat anti-rabbit IgG | JIR: Cat# 111-035-047 | RRID:AB_2337940 | WB (1:5000) |
| Chemical compound, drug | Tamoxifen | Sigma | Cat# T5648 | |
| Chemical compound, drug | Paraformaldehyde (PFA) | Sigma | Cat# P6148-500g | |
| Chemical compound, drug | Triton X-100 | Sigma | Cat# T9284 | |
| Chemical compound, drug | 4'6-Diamidino-2-phenylindole (DAPI) | Vector Lab | Cat# D21490 | |
| Chemical compound, drug | Collagenase type I | Gibco | Cat# 17100-017 | |
| Chemical compound, drug | Percoll | GE Healthcare | Cat# 17-0891-01 | |
| Chemical compound, drug | DNase I | Worthington | Cat# LS002139 | |
| Chemical compound, drug | TRIzol | Invitrogen | Cat# 15596018 | |
| Chemical compound, drug | RIPA lysis buffer | Beyotime | Cat# P0013B | |
| Chemical compound, drug | Protease inhibitors | Roche | Cat# 11836153001 | |
| Chemical compound, drug | 5× loading buffer | Beyotime | Cat# P0015L | |
| Chemical compound, drug | Precast gradient gels | Beyotime | Cat# P0469M | |

*Appendix 1 Continued on next page*

*Appendix 1 Continued*

| Reagent type (species) or resource | Designation | Source or reference | Identifiers | Additional information |
|---|---|---|---|---|
| Chemical compound, drug | Immobilon PVDF membranes | Millipore | Cat# IPVH00010 | |
| Chemical compound, drug | 1× running buffer | Epizyme | Cat# PS105S | |
| Chemical compound, drug | Blocking buffer | Epizyme | Cat# PS108P | |
| Chemical compound, drug | Primary antibody dilution buffer | Epizyme | Cat# PS114 | |
| Chemical compound, drug | 1× TBST | Epizyme | Cat# PS103S | |
| Commercial assay or kit | Lipofectamine 3000 Transfection Reagent | Thermo Fisher | Cat# L3000015 | |
| Commercial assay or kit | Prime Script RT kit | Takara | Cat# RR047A | |
| Commercial assay or kit | SYBR Green qPCR master mix | Thermo Fisher | Cat# 4367659 | |
| Commercial assay or kit | Pierce BCA Protein Assay kits | Thermo Scientific | Cat# 23227 | |
| Commercial assay or kit | Chemiluminescent HRP substrate | Thermo Fisher | Cat# WBKLS0500 | |
| Software, algorithm | Fiji | Version: 2.9.0/1.54f | RRID:SCR_002285 | https://imagej.net/software/fiji/ |
| Software, algorithm | GraphPad Prism | Version: 9.5.1 | RRID:SCR_002798 | https://www.graphpad.com |
| Software, algorithm | FlowJo | Version: 10.9.0 | RRID:SCR_008520 | https://www.flowjo.com |
| Software, algorithm | Photoline | Version: 23.02 | RRID:SCR_027878 | http://pl64.com |
| Other | Olympus microscope BX53 | Olympus | | Used for image acquisition after immunofluorescence staining |
| Other | Olympus confocal FV3000 | Olympus | | Used for image acquisition after immunofluorescence staining |
| Other | Zeiss stereoscopic microscope AxioZoom V16 | Zeiss | | Used for whole-mount image acquisition |
| Other | Nikon A1 FLIM | Nikon | | Used for image acquisition after immunofluorescence staining |
| Other | Leica SP8 WILL | Leica | | Used for image acquisition after immunofluorescence staining |
| Other | QuantStudio 6 Real-Time PCR System | Thermo Fisher | | Used for the quantitative detection of qPCR |
| Other | MiniChemi 610 Plus | Biogamma | | Used for acquiring western blot results |
| Other | Beckman Coulter CytoFLEX LX 5-Laser Stem Cell Multicolor Flow Cytometer | Beckman | | Used for flow cytometry analysis |
| Other | FACSAria SORP | Becton, Dickinson and Company | | Used for flow cytometry sorting |
| Other | Sony MA900 | Sony | | Used for flow cytometry sorting |

