## [Editor Report · eLife Assessment]

This study presents an **important** set of new tools to facilitate Cre or Dre-mediated recombination in mice. The characterization of these new tools was done using **solid** and validated methodology. The work **convincingly** demonstrates the efficient gene knockout capability of these models and will progress the field.

---

## [Referee Report · Reviewer #1 (Public review)]

This is a simple and potentially valuable approach to reduce Cre leak in amplified systems designed to improve CreER use across alleles. The revised work is improved with a direct comparison to the Benedito iSure-Cre line, providing some practical guidance for investigators. The authors do not address the issue of Cre toxicity or mosaic efficiency with low Tamoxifen use.

The major improvement in my mind is the inclusion of Supp Fig 7 where the authors compare their loxCre to iSureCre. The discussion is somewhat improved, but still fails to discuss significant issues such as Cre toxicity in detail. As noted by most reviewers, without a biological question, the paper is entirely a technical description of a couple of new tools. Whether and to what extent journals such as eLife should publish every new technical innovation without rigorous functional comparison to prior tools is an important question raised by this study. There is already a plethora of available techniques, most of which look better on paper than they function in mice.

However, I do feel that these tools will be of potential use to the field.

---

## [Referee Report · Reviewer #2 (Public review)]

This work presents new genetic tools for enhanced Cre-mediated gene deletion and genetic lineage tracing. The authors optimise and generate mouse models that convert temporally controlled CreER or DreER activity to constitutive Cre expression, coupled with the expression of tdT reporter for the visualizing and tracing of gene-deleted cells. This was achieved by inserting a stop cassette into the coding region of Cre, splitting it into N- and C-terminal segments. Removal of the stop cassette by Cre-lox or Dre-rox recombination results in the generation of modified Cre that is shown to exhibit similar activity to native Cre. The authors further demonstrate efficient gene knockout in cells marked by the reporter using these tools, including intersectional genetic targeting of pericentral hepatocytes.

The new models offer several important advantages. They enable tightly controlled and highly effective genetic deletion of even alleles that are difficult to recombine. By coupling Cre expression to reporter expression, these models reliably report Cre-expressing i.e. gene-targeted cells and circumvent false positives that can complicate analyses in genetic mutants relying on separate reporter alleles. Moreover, the combinatorial use of Dre/Cre permits intersectional genetic targeting, allowing for more precise fate mapping.

The study and the new models have also limitations. The demonstration of efficient deletion of multiple floxed alleles in a mosaic fashion, a scenario where the lines would demonstrate their full potential compared to already existing models, has not been tested in the current study. Mosaic genetics is increasingly recognized as a key methodology for assessing cell-autonomous gene functions. The challenge lies in performing such experiments, as low doses of tamoxifen needed for inducing mosaic gene deletion may not be sufficient to efficiently recombine multiple alleles in individual cells while at the same time accurately reporting gene deletion. In addition, as discussed by the authors, a limitation of this line is the constitutive expression of Cre, which is associated with toxicity in some cases.

Comments on revisions: I have no further comments.

---

## [Referee Report · Reviewer #3 (Public review)]

Shi et al describe a new set of tools to facilitate Cre or Dre-recombinase-mediated recombination in mice. The strategies are not completely novel but have been pursued previously by the lab, which is world-leading in this field, and by others. The authors report a new version of the iSuRe-Cre approach, which was originally developed by Rui Benedito's group in Spain. Shi et al describe that their approach shows reduced leakiness compared to the iSuRe-Cre line. Furthermore, a new R26-roxCre-tdT mouse line was established after extensive testing, which enables efficient expression of the Cre recombinase after activation of the Dre recombinase. The authors carefully evaluated efficiency and leakiness of the new line and demonstrated the applicability by marking peri-central hepatocytes in an intersectional genetics approach. The paper represents the result of enormous, carefully executed efforts. Although I would have preferred to see a study which uses the wonderful new tools to address a major biological question, carefully conducted technical studies have an enormous value for the scientific community, clearly justifying publication.

The new mouse lines generated in this study will enhance the precision of genetic manipulation in distinct cell types and greatly facilitate future work in numerous laboratories. The authors expertly eradicated weaknesses from initial submissions. Remaining open questions regarding potential toxicity of expressing multiple recombinases and fluorescence reports were convincingly answered.

---

## [Author Response]

The following is the authors’ response to the previous reviews

**Public Reviews:**

**Reviewer #1 (Public Review):**
(1) It is a nice study but lacks some functional data required to determine how useful these alleles will be in practice, especially in comparison with the figure line that stimulated their creation.

We are grateful for this comment. For the usefulness of these alleles, Figure 3 shows that specific and efficient genetic manipulation of one cell subpopulation can be achieved by mating across the *DreER* mouse strain to the *rox-Cre* mouse strain. In addition, Figure 7 shows that *R26-loxCre-tdT* can effectively ensure Cre-loxP recombination on some gene alleles and for genetic manipulation. The expression of the tdT protein is aligned with the expression of the Cre protein (*Alb roxCre-tdT* and *R26-loxCre-tdT*, Figure 2 and Figure 5), which ensures the accuracy of the tracing experiments. We believe more functional data can be shown in future articles that use mice lines mentioned in this manuscript.

(2) The data in Figure 5 show strong activity at the Confetti locus, but the design of the newly reported R26-loxCre line lacks a WPRE sequence that was included in the iSure-Cre line to drive very robust protein expression.

Thank you for bringing up this point in the manuscript. In the *R26-loxCre-tdT* mice knock-in strategy, the WPRE sequence is added behind the *loxCre-P2A-tdT* sequence, as shown in Supplementary Figure 9.

(3) The most valuable experiment for such a new tool would be a head-to-head comparison with iSure (or the latest iSure version from the Benedito lab) using the same CreER and target foxed allele. At the very least a comparison of Cre protein expression between the two lines using identical CreER activators is needed.

Thank you for your valuable and insightful comment. The comparison results of *R26-loxCre-tdT* with *iSuRe-Cre* using *Alb-CreER* and targeting *R26-Confetti* can be found in Figure 6, according to the reviewer’s suggestion.

(4) Why did the authors not use the same driver to compare mCre 1, 4, 7, and 10? The study in Figure 2 uses Alb-roxCre for 1 and 7 and Cdh5-roxCre for 4 and 10, with clearly different levels of activity driven by the two alleles in vivo. Thus whether mCre1 is really better than mCre4 or 10 is not clear.

Response: or two mCre versions that work efficiently. For example, if *Alb-mCre1* was competitive with *Cdh5-mCre10*, we can use them for targeting genes in different cell types, broadening the potential utility of these mice.

(5) Technical details are lacking. The authors provide little specific information regarding the precise way that the new alleles were generated, i.e. exactly what nucleotide sites were used and what the sequence of the introduced transgenes is. Such valuable information must be gleaned from schematic diagrams that are insufficient to fully explain the approach.

Response: We appreciate your thoughtful suggestions. The schematic figures, along with the nucleotide sequences for the generation of mice, can be found in the revised Supplementary Figure 9.

**Reviewer #2 (Public Review):**
(1) The scenario where the lines would demonstrate their full potential compared to existing models has not been tested.

Thank you for your thoughtful and constructive comment. The comparative analysis of *R26-loxCre-tdT* with *iSuRe-Cre*, employing *Alb-CreER* to target *R26-Confetti*, is provided in Figure 6.

(2) The challenge lies in performing such experiments, as low doses of tamoxifen needed for inducing mosaic gene deletion may not be sufficient to efficiently recombine multiple alleles in individual cells while at the same time accurately reporting gene deletion. Therefore, a demonstration of the efficient deletion of multiple floxed alleles in a mosaic fashion would be a valuable addition.

Thank you for your constructive comments. Mosaic analysis using sparse labeling and efficient gene deletion would be our future direction using roxCre and loxCre strategies.

1. When combined with the confetti line, the reporter cassette will continue flipping, potentially leading to misleading lineage tracing results.

Thank you for your professional comments. Indeed, the confetti used in this study can continue flipping, which would lead to potentially misleading lineage tracing results. Our use of *R26-Confetti* is to demonstrate the robustness of mCre for recombination. Some multiple-color mice lines that don’t flip have been published, for example, R26-Confetti2(PMID: 30778223) and Rainbow (PMID: 32794408). These reporters could be used for tracing Cre-expressing cells, without concerns of flipping of reporter cassettes.

(4) Constitutive expression of Cre is also associated with toxicity, as discussed by the authors in the introduction.

Thank you for your professional comments. The toxicity of constitutive expression of Cre and the toxicity associated with tamoxifen treatment in CreER mice line (PMID: 37692772) are known to the field. This study can’t solve the toxicity of the constitutive expression of Cre in this work. Many mouse lines with constitutive Cre driven by different promoters are present across various fields, representing similar toxicity. To solve this issue, it would be possible to construct a new strategy that enables the removal of Cre after its expression.

**Reviewer #3 (Public Review):**
(1) Although leakiness is rather minor according to the original publication and the senior author of the study wrote in a review a few years ago that there is no leakiness(https://doi.org/10.1016/j.jbc.2021.100509).

Thank you so much for your careful check. In this review (PMID: 33676891), the writer’s comments on *iSuRe-Cre* are on the reader's side, and all summary words are based on the original published paper (PMID: 31118412). Currently, we have tested *iSuRe-Cre* in our hands. We did detect some leakiness in the heart and muscle, but hardly in other tissues as shown in Author response image 1.

**Author response image 1. sa4fig1:** Leakiness in *Alb CreER*;*iSuRe-Cre* mouse line. Pictures are representative results for 5 mice. Scale bars, white 100 µm.

(2) I would have preferred to see a study, which uses the wonderful new tools to address a major biological question, rather than a primarily technical report, which describes the ongoing efforts to further improve Cre and Dre recombinase-mediated recombination.

Response: We gratefully appreciate your valuable comment. The roxCre and loxCre mice mentioned in this study provide more effective methods for inducible genetic manipulation in studying gene function. We hope that the application of our new genetic tools could help address some major biological questions in different biomedical fields in the future.

(3) Very high levels of Cre expression may cause toxic effects as previously reported for the hearts of Myh6-Cre mice. Thus, it seems sensible to test for unspecific toxic effects, which may be done by bulk RNA-seq analysis, cell viability, and cell proliferation assays. It should also be analyzed whether the combination of R26-roxCre-tdT with the Tnni3-Dre allele causes cardiac dysfunction, although such dysfunctions should be apparent from potential changes in gene expression.

We are sorry that we mistakenly spelled *R26-loxCre-tdT* into *R26-roxCre-tdT* in our manuscript. We have not generated the *R26-roxCre-tdT* mouse line. We also thank the reviewer for concerns about the toxicity of high Cre expression. The toxicity of constitutive expression of Cre and the toxicity of tamoxifen treatment of CreER mice line (PMID: 37692772) are known to the field. This study can’t solve the toxicity of the constitutive expression of Cre in this work. Many mouse lines with constitutive Cre driven by different promoters are present across various fields, representing similar toxicity. To solve this issue, it would be possible to construct a new strategy that enables the removal of Cre after its expression.

(4) Is there any leakiness when the inducible DreER allele is introduced but no tamoxifen treatment is applied? This should be documented. The same also applies to loxCre mice.

In this study, we come up with new mice tool lines, including *Alb roxCre1-tdT*, *Cdh5 roxCre4-tdT*, *Alb roxCre7-GFP*, *Cdh5 roxCre10-GFP* and *R26-loxCre-tdT*. As the data shown in Supplementary Figure 1, Supplementary Figure 2, and Figure 4D, *Alb roxCre1-tdT*, *Cdh5 roxCre4-tdT*, *Alb roxCre7-GFP*, *Cdh5 roxCre10-GFP* and *R26-loxCre-tdT* are not leaky. Therefore, if there is any leakiness driven by the inducible DreER or CreER allele, the leakiness is derived from the DreER or CreER. Additional pertinent experimental data can be referenced in Figure S4C, Figure S7A-B, and Figure S8A.

(5) It would be very helpful to include a dose-response curve for determining the minimum dosage required in Alb-CreER; R26-loxCre-tdT; Ctnnb1flox/flox mice for efficient recombination.

Thank you for your suggestion. We value your feedback and have incorporated your suggestion to strengthen our study. Relevant experimental data can be referenced in Figure S8E-G.

(6) In the liver panel of Figure 4F, tdT signals do not seem to colocalize with the VE-cad signals, which is odd. Is there any compelling explanation?

The staining in Figure 4F in the revision is intended to deliver optimized and high-resolution images.

(7) The authors claim that "virtually all tdT+ endothelial cells simultaneously expressed YFP/mCFP" (right panel of Figure 5D). Well, it seems that the abundance of tdT is much lower compared to YFP/mCFP. If the recombination of R26-Confetti was mainly triggered by R26-loxCre-tdT, the expression of tdT and YFP/mCFP should be comparable. This should be clarified.

Thank you so much for your careful check. We checked these signals carefully and didn't find the “much lower” tdT signal. As the file-loading website has a file size limitation, the compressed image results in some signal unclear. We attached clear high-resolution images here. Author response image 2 shows how we split the tdT signal and compared it with YFP/mCFP.

**Author response image 2. sa4fig2:** 

(8) In several cases, the authors seem to have mixed up "R26-roxCre-tdT" with "R26-loxCre-tdT". There are errors in #251 and #256.Furthermore, in the passage from line #278 to #301. In the lines #297 and #300 it should probably read "Alb-CreER; R26-loxCretdT; Ctnnb1flox/flox" rather than "Alb-CreER;R26-tdT2;Ctnnb1flox/flox".

We are grateful for these careful observations. We have corrected these typos accordingly.

**Recommendations for the authors:**

**Reviewer #1:**
(1) However, for it to be useful to investigators a more direct comparison with the Benedito iSure line (or the latest version) is required as that is the crux of the study.

Thank you for emphasizing this point, which we have now addressed in the revised manuscript and Figure 6.

(2) I would like to know how the authors will make these new lines available to outside investigators.

Please contact the lead author by email to consult about the availability of new mouse lines developed in this study.

(3) The discussion is overly long and fails to address potential weaknesses. Much of it reiterates what was already said in the results section.

We are thankful for your critical evaluation, which has helped us improve our discussion.

**Reviewer #2:**
(1) Assessing the efficiency and accuracy of the lines in mosaic deletions of multiple alleles and reporting them in single cells after low-dose tamoxifen exposure would be highly beneficial to demonstrate the full potential of the models.

We appreciate your careful consideration of this issue. Our future endeavors will focus on mosaic analysis utilizing sparse labeling and efficient gene deletion, employing both roxCre and loxCre strategies.

(2) Performing FACS analysis to confirm that all targeted (Cre reporter-positive) cells are also tdT-positive would provide more precise data and avoid vague statements like 'virtually all' or 'almost complete' in the results section:Line 166: Although mCre efficiently labeled virtually all targeted cells (Figure S3A-E)...Line 293: ... and not a single tdT+ hepatocyte 293 expressed Cyp2e1 (Figure 6D)... However, the authors do not provide any quantification. FACS would be ideal here.Line 244: ...expression of beta-catenin and GS almost disappeared in the 4W mutant sample... The resolution in the provided PDF is not adequate for assessment.Line 296: ... revealed almost complete deletion of Ctnnb1 in the Alb-CreER;R26-tdT2;Ctnnb1flox/flox mice...

Thank you for suggesting these improvements, which have strengthened the robustness of our conclusions. In the revised version, we have incorporated FACS results that correspond to related sections. Additionally, a quantification statement has been included in the statistical analysis section. We appreciate your meticulous review and comments, which have significantly improved the clarity of our manuscript.

(3) In the beginning of the results section, it is not clear which results are from this study and which are known background information (like Figure 1A). For example, it is not clear if Figure 1C presents data from R26-iSuRe-Cre. Please revise the text to more clearly present the experimental details and new findings.

Thank you for this observation. Figure 1C belongs to this study, and the revised version has been modified to the related statement for improved clarity.

(4) Experimental details regarding the genetic constructs and genotyping of the new knock-in lines are missing. Are R26 constructs driven by the endogenous R26 promoter or were additional enhancers used?

Thank you for emphasizing this point. The schematic figures and nucleotide sequences for the generation of mice can be found in the revised Supplementary Figure 9, which can help to address this issue.

(5) The method used to quantify mCre activity in terms of reporter+ target cells is not specified. From images or by FACS?Additionally, if images were used for quantification, it would be important to provide details on the number of images analyzed, the number of cells counted per image, and how individual cells were identified.

Thank you for your comment. We have included the quantification statement in the statistical analysis section. Analyzing *R26-Confetti*+ target cells using FACS is challenging due to the limitations of the sorting instrument. Consequently, we quantified the related data by images. Each dot on the chart represents one sample, and the quantification for each mouse was conducted by averaging the data from five 10x fields taken from different sections.

(6) Line 160: These data demonstrate that roxCre was functionally efficient yet non-leaky. Functional efficiency in vivo was not shown in the preceding experiments.

Functional efficiency in vivo can be referred to in Figures S1-S2 and S4C.

(7) It would be useful to provide a reference for easy vs low-efficiency recombination of different reporter alleles (lines 56-58).

We are grateful for this comment, as it has allowed us to improve the clarity of our explanation. Consequently, we have made the necessary modifications.

(8) Discussion on the potential drawbacks and limitations of the lines would be useful.

We are thankful for your evaluation, which has significantly contributed to the enhancement of our discourse.